# Myeloid cell nuclear differentiation antigen controls the pathogen-stimulated type I interferon cascade in human monocytes by transcriptional regulation of IRF7

Lili Gu[1,6], David Casserly[1,6], Gareth Brady[2], Susan Carpenter[3], Adrian P. Bracken[4], Katherine A. Fitzgerald [3], Leonie Unterholzner [1,5] & Andrew G. Bowie [1✉]

Type I interferons (IFNs) are critical for anti-viral responses, and also drive autoimmunity when dysregulated. Upon viral sensing, monocytes elicit a sequential cascade of IFNβ and IFNα production involving feedback amplification, but how exactly this cascade is regulated in human cells is incompletely understood. Here we show that the PYHIN protein myeloid cell nuclear differentiation antigen (MNDA) is required for IFNα induction in monocytes. Unlike other PYHINs, this is not due to a pathogen sensing role, but rather MNDA regulated expression of IRF7, a transcription factor essential for IFNα induction. Mechanistically, MNDA is required for recruitment of STAT2 and RNA polymerase II to the IRF7 gene promoter, and in fact MNDA is itself recruited to the IRF7 promoter after type I IFN stimulation. These data implicate MNDA as a critical regulator of the type I IFN cascade in human myeloid cells and reveal a new role for human PYHINs in innate immune gene induction.

[1] School of Biochemistry and Immunology, Trinity Biomedical Sciences Institute, Trinity College Dublin, Dublin 2, Ireland. [2] School of Medicine, Trinity College Dublin, Dublin 2, Ireland. [3] Division of Innate Immunity, University of Massachusetts Chan Medical School, Worcester, MA, USA. [4] Smurfit Institute of Genetics, Trinity College Dublin, Dublin 2, Ireland. [5] Present address: Division of Biomedical and Life Sciences, Faculty of Health and Medicine, Lancaster University, Lancaster LA1 4YQ, UK. [6] These authors contributed equally: Lili Gu, David Casserly. ✉email: agbowie@tcd.ie

Type I interferons (IFNs) have essential roles in regulating immune and inflammatory responses. Depending on the context they can be both protective or pathogenic: as well as being the critical components of the early anti-viral innate immune response, and regulators of adaptive immunity, they can also initiate or sustain autoimmune diseases[1]. In fact, several autoimmune conditions are now defined as 'interferonopathies', recognising the role of dysregulated type I IFN in disease pathology[2]. Therefore, it is important to more fully understand the mechanisms whereby type I IFNs are regulated. Further, our interest here is in focusing on the human system, since differences can exist in how type I IFNs operate and are induced between humans and oft used mouse models of IFN activity and function[3]. Two main sub-groups of type I IFNs that are induced in response to detection of pathogen-associated molecular patterns (PAMPs) from viruses and also to danger signals such as mislocalised nucleic acid are IFNβ and IFNα subtypes[4].

Cell surface and endosomal pattern recognition receptors (PRRs) especially Toll-like receptors (TLRs) can detect the presence of viral and mislocalised nucleic acid, and via the signalling proteins MyD88 and Toll-interleukin-1 (IL-1) receptor domain-containing adaptor inducing IFNβ (TRIF) activate the TANK binding kinase-1-IFN regulatory factor 3 (TBK1-IRF3) signalling axis, that causes rapid IFNβ gene induction[5]. The TBK1-IRF3-IFNβ axis is also activated by intracellular PRRs. RIG-I-like receptors (RLRs) sense RNA from intracellular viruses, and engage TBK1 via mitochondrial antiviral signalling protein (MAVS), while intracellular dsDNA sensors such a cGAMP synthase (cGAS) and the PYHIN (Pyrin and HIN domain) protein IFI16 (IFN-γ inducible protein 16) detect viral and mislocalised DNA and also activate TBK1-IRF3-IFNβ via the adaptor STING (stimulator of IFN genes)[6,7]. PRR activation then triggers a type I IFN cascade whereby IFNβ released from the sensing cell binds to the IFN-α/β receptor (IFNAR) on sensing and surrounding cells. Engagement of IFNβ with IFNAR activates a JAK (Janus kinase)-STAT (signal transducer and activator of transcription) signalling cascade, leading to phosphorylation and activation of STAT1 and STAT2. STAT1 and STAT2 then form a complex with IRF9 termed the IFN-stimulated gene factor 3 (ISGF3) which binds to and induces the IFN-stimulated response element (ISRE) in the gene promoters of ISGs, including *IRF7*[8]. IRF7 is the key transcription factor which induces IFNα expression[9], leading to a second wave of type I IFN release. Since IFNα signals via the IFNAR this leads to a positive feedback loop that magnifies the type I IFN response by amplifying IRF7 and IFNα induction.

As mentioned above, IFI16 is a human PYHIN protein that can sense the presence of intracellular pathogen dsDNA leading to type I IFN induction[10–12]. PYHIN proteins are defined in most cases by the presence of a HIN200 motif that can bind dsDNA, and a pyrin domain that can mediate protein–protein interactions[13]. IFI16 also activates a STING-dependent signalling pathway in response to sensing of DNA damage in the nucleus[14]. A further role for IFI16 that has emerged in innate immunity is as a viral restriction factor, in that IFI16 can directly target viral genomes or sequester transcription factors required by viruses and thus inhibit their replication and transcription. This is true for herpesviruses including cytomegalovirus (CMV)[15–17], for human papillomavirus[18] and for lentiviruses including HIV-1[19]. Apart from IFI16, there are four other human PYHIN proteins, namely absent in melanoma 2 (AIM2), PYHIN family member 1 (PYHIN1, also called IFIX), myeloid cell nuclear differentiation antigen (MNDA) and pyrin only protein 3 (POP3)[13]. AIM2 engages dsDNA and forms an inflammasome leading to caspase 1 activation and thus pyroptosis and IL-1β release[20,21]. POP3 is a pyrin-only protein

which can negatively regulate AIM2 activity[22]. Like IFI16, PYHIN1 and MNDA have recently been shown to also restrict HIV-1, via a shared mechanism of sequestering the transcription factor Sp1 away from HIV-1 gene promoters[23]. PYHIN1 also restricts herpesviruses[24] and may also have a role as a dsDNA PRR[25], while we recently showed a further role for PYHIN1 in innate immunity in that it regulates pro-inflammatory cytokine production in human airway epithelial cells[26]. MNDA remains the least well characterised human PYHIN protein in terms of potential roles in innate immunity.

Although numerous cell types can secrete IFNβ in response to viral infection, myeloid cells such as monocytes and plasmacytoid dendritic cells (pDCs) are noteworthy in the quantity of type I IFN they produce during an infection, and also since they are particularly implicated in interferonopthies[1]. Given the important role of the type I IFN cascade in human myeloid cells, and the fact that MNDA is a myeloid-specific PYHIN protein[27], we examined a potential role for MNDA in pathogen sensing and IFN induction in myeloid cells. MNDA has previously been implicated in myeloid cell differentiation[28,29] and in neutrophil apoptosis[30]. Here we reveal that MNDA is a critical regulator of the type I IFN cascade in myeloid cells. Unlike IFI16, MNDA does not sense dsDNA, but rather is required by all inducers of type I IFN due to a role in IRF7 induction. Surprisingly, MNDA is required for enhanceosome formation on the human *IRF7* promoter and is itself recruited to the *IRF7* promoter in response to IFNAR stimulation. Thus, MNDA is a newly revealed critical regulator of the type I IFN cascade in human myeloid cells.

## Results

**MNDA regulates dsDNA-stimulated IFNα induction in human monocytes.** To study the function of MNDA in innate immunity, we first confirmed which cell types MNDA protein was expressed in, in human blood. For this, peripheral blood mononuclear cells (PBMCs) isolated from buffy coats were stained with an anti-MNDA antibody together with antibodies for cell-specific markers, and then analysed by flow cytometry. Cells were gated on lymphocytes for the lymphoid markers CD3 (for T cells), CD19 (for B cells) and CD56 (for NK cells), and the non-lymphocyte cells were gated for the myeloid marker CD14 (for monocytes) (Supplementary Fig. 1). Consistent with the human proteome map[31], in a mixed population of PBMCs MNDA was found to be primarily expressed in CD14+ monocytes and CD19+ B cells (Supplementary Fig. 1). Immunoblot analysis confirmed strong expression of MNDA in monocytes, which was reduced upon differentiation of cells into different macrophage lineages by treatment of monocytes with either GM-CSF or M-CSF[32] (Fig. 1a). The human monocytic cell line THP-1 displayed a similar profile of MNDA protein expression to primary human monocytes in that undifferentiated cells showed higher protein expression of MNDA than cells differentiated into macrophages using phorbol-12-myristate-13-acetate (PMA), and IFNα or IFNγ had little effect on protein expression in undifferentiated cells (Fig. 1b). For comparison, we also measured expression of the whole PYHIN family in THP-1s. Both mRNA (Supplementary Fig. 2a–d) and protein (Fig. 1b) expression of the four main human PYHINs was evident in THP-1 cells, although AIM2 was only detected after IFNγ treatment. These data established THP-1 cells as a good model system to examine MNDA function.

Since MNDA protein expression was higher in unstimulated monocytes than macrophages, in contrast to IFI16 where the opposite was the case (Fig. 1b), we first wondered whether MNDA in monocytes fulfilled a function similar to IFI16 in macrophages as a cytosolic DNA sensor[10]. If this were the case then MNDA would be expected to be required for

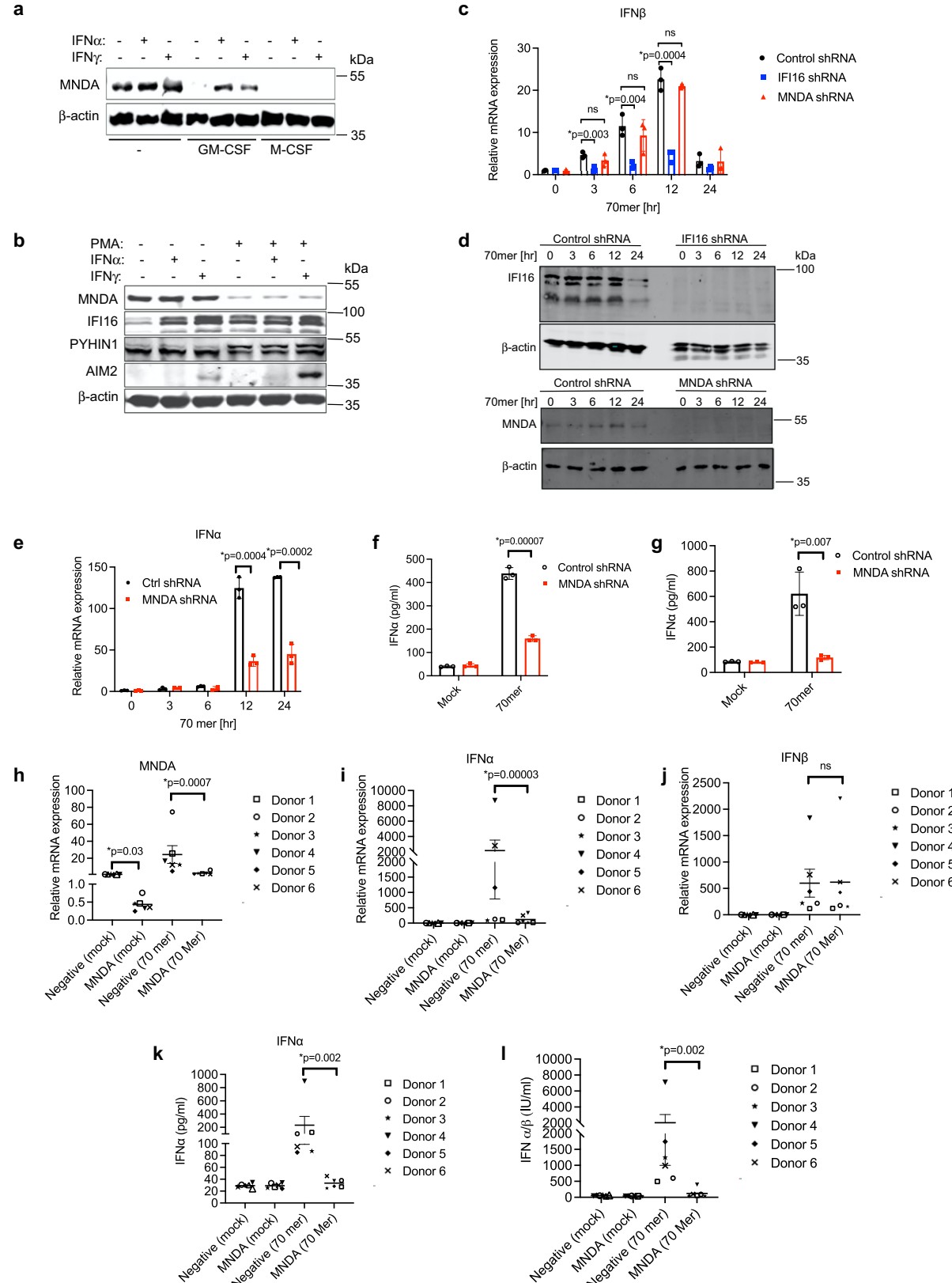

dsDNA-stimulated early IFNβ induction, since this type I IFN is induced directly after cytosolic DNA sensing via activation of the STING-TBK1-IRF3 signalling axis[33]. To test this, we generated THP-1s stably expressing MNDA shRNA, which displayed substantially reduced MNDA protein expression (Fig. 1d), for loss of function studies. However, stimulation of IFNβ mRNA

induction via cytosolic DNA sensing activated by transfection of dsVACV DNA (a 70 nt long immunostimulatory dsDNA derived from vaccinia virus[10]) was not impaired in THP-1 cells expressing MNDA shRNA (Fig. 1c). This contrasted with the situation in THP-1 cells expressing IFI16 shRNA, which showed impaired IFNβ mRNA induction after dsDNA treatment

**Fig. 1 MNDA regulates dsDNA-stimulated IFNα induction in human monocytes. a** Immunoblot analysis of MNDA in primary monocytes treated with IFNα (1000 U/ml) or IFNγ (50 ng/ml) for 24 h or grown in the presence of GM-CSF or M-CSF for 7 days and treated with IFNα or IFNγ for 24 h. Representative of three experiments. **b** Immunoblot analysis of MNDA, IFI16, PYHIN1 and AIM2 in unprimed or PMA primed THP-1 cells treated with IFNα (1000 U/ml) or IFNγ (50 ng/ml) for 24 h. Representative of three experiments. **c** Quantitative PCR analysis of IFNβ mRNA from THP-1 cells expressing control and MNDA or IFI16 shRNA transfected with dsVACV 70mer (1 μg/ml) for the times indicated. **d** Western blot confirming knockdown of IFI16 and MNDA expression in shRNA cells. **e** Quantitative PCR analysis of IFNα mRNA from THP-1 cells expressing control or MNDA shRNA transfected with dsVACV 70mer (1 μg/ml) for the indicated times. **f, g** Release of IFNα protein from unprimed (**f**) or PMA-primed (**g**) THP-1 cells expressing control or MNDA shRNA, transfected with dsVACV 70mer (1 μg/ml) for 24 h. **h–j** Quantitative PCR analysis of MNDA (**h**), IFNα (**i**) and IFNβ (**j**) mRNA from primary human blood monocytes electroporated with control or MNDA siRNA and transfected with dsVACV 70mer (1 μg/ml) for 24 h. **k** Secreted IFNα from primary human blood monocytes electroporated with control or MNDA siRNA and transfected with dsVACV 70mer (1 μg/ml) for 24 h was measured by ELISA. **l** IFNα/β bioactivity in supernatants from primary human blood monocytes electroporated with control or MNDA siRNA and transfected with dsVACV 70mer (1 μg/ml) for 24 h was measured by Bioassay. For (**h–k**), data shown is mean ± SD from six donors (each data point is a single donor). All other data are mean ± SD of triplicate samples and are representative of three independent experiments; two tailed unpaired Students t test; *p < 0.05 indicates significance compared to respective groups; ns indicates not significant.

(Fig. 1c, d). Thus, unlike IFI16, MNDA does not function as a sensor of dsDNA and may have a distinct function from IFI16 in monocytic cells.

As well as assessing IFNβ mRNA induction, we also examined IFNα mRNA after dsDNA stimulation of cells. To our surprise, although IFNβ mRNA induction was not MNDA-dependent, IFNα mRNA induction was significantly impaired in THP-1 cells expressing MNDA shRNA (Fig. 1e), as was IFNα protein release from cells (Fig. 1f). The requirement for MNDA for IFNα production was also seen in differentiated monocytes (Fig. 1g), even though these cells express less MNDA than monocytes (Fig. 1b). Further, reduction of MNDA expression in primary human blood monocytes, by transient transfection of siRNA oligonucleotides targeting MNDA sequences distinct from the shRNA constructs used in THP-1 cells (Fig. 1h), led to significant inhibition of DNA-stimulated IFNα mRNA induction (Fig. 1i), but not IFNβ mRNA induction (Fig. 1j). This translated to a significant reduction of released IFNα, (Fig. 1k) and thus an overall reduction of type I IFN as measured by bioassay (Fig. 1l). Together these results demonstrate that MNDA is required for IFNα mRNA induction in human monocytes and macrophages.

**MNDA regulates dsRNA- and virus-stimulated IFNα induction in human monocytes.** As well as dsDNA, dsRNA is a potent inducer of IFNα so we next determined whether MNDA was also required for this response. Similar to the case for dsDNA, transfection of monocytes with dsRNA led to MNDA-dependent IFNα induction (Fig. 2a), while IFNβ was unaffected (Fig. 2b). This was also true for dsRNA stimulation when MNDA expression was supressed using transient siRNA to target a heterogeneous population of THP-1 cells (Supplementary Fig. 3a). In that case, dsRNA-stimulated IFNα mRNA induction was potently supressed (Supplementary Fig. 3b), while a minor effect on IFNβ was also observed (Supplementary Fig. 3c), likely due to secondary IFNα-dependent IFNβ induction (see below). We also infected THP-1 cells with live RNA viruses to assess a role for MNDA in virus-induced IFNα. Thus THP-1 cells were infected with negative-stranded RNA viruses, either Sendai virus (SeV) or vesicular stomatitis virus (VSV). This showed that MNDA was required for RNA virus-stimulated IFNα protein secretion from monocytes for both types of viruses (Fig. 2c, d). Since VSV expressed GFP, it was also possible to determine the effect of supressed MNDA expression on VSV replication, by measuring GFP expression in cells, and this showed significantly enhanced GFP expression, and thus increased VSV replication, in cells with reduced MNDA expression (Fig. 2e, Supplementary Fig 4). These data suggest that MNDA has an intrinsic role in IFNα induction in monocytes that is independent of the stimulus used (dsDNA, dsRNA, RNA virus).

**MNDA does not regulate PRR or IFNAR signalling.** To investigate why MNDA was required for IFNα induction we considered how nucleic acids and viruses cause IFNα induction in monocytes. Figure 3a outlines the type I IFN cascade that is triggered by PRR sensing of nucleic acids and viruses: these PRRs signal via adaptor proteins such as STING, MAVS and TRIF to activate TBK1, which phosphorylates IRF3, the main transcription factor that needs to be activated to induce IFNβ expression. Secreted IFNβ protein then signals via the IFNα/β receptor (IFNAR) leading to STAT1 and STAT2 activation, subsequent induction of ISGs including IRF7 and eventual expression of IFNα (Fig. 3a). Consistent with the lack of requirement of MNDA for IFNβ induction, DNA-stimulated IRF3 activation, as measured by the appearance of phosphorylated IRF3 on a Western blot, was normal in THP-1 cells expressing MNDA shRNA compared to control shRNA cells (Supplementary Fig. 5a). A role for MNDA in IFNα induction could be explained by a requirement for MNDA in the IFNAR signalling pathway leading to ISG induction. However, DNA-stimulated STAT1 activation was also normal in cells expressing MNDA shRNA (Supplementary Fig. 5a), as was DNA-stimulated induction of the ISGs IFIT1, IFIT2 and IFIT3 (Supplementary Fig. 5b–d). Consistent with these data, STAT1 activation and induction of the ISGs IFIT3, IFIT1, IFIT2, IRF1 and ISG15 by direct stimulation of the IFNAR by addition of type I IFN to cells, were also MNDA-independent (Supplementary Fig. 5e–i). Hence the requirement for MNDA in nucleic acid-stimulated IFNα induction could not be explained by a role for MNDA in PRR or IFNAR signalling.

**MNDA controls IRF7-dependent gene induction.** To further explore the role of MNDA in IFNα induction we used CRISPR/Cas9 to delete *MNDA* from THP-1 cells. Two different guide RNAs targeting *MNDA* were delivered to THP-1 cells stably expressing Cas9[34,35] (Supplementary Fig. 6a). We selected three clones confirmed to have a disrupted *MNDA* gene locus (Supplementary Fig. 6b–d), which each showed no MNDA protein expression in the presence or absence of IFNα compared to three control clones (Fig. 4a).

Gene deletion of MNDA showed very significant inhibition of DNA-stimulated IFNα induction (Fig. 3b). Therefore, we next looked at specific induction of IFNα gene promoters in more detail, by using chromatin immunoprecipitation (ChIP) to measure recruitment of transcription factors and RNA polymerase II (Pol II) to type I IFN promoters. The human IFNα gene family includes 13 distinct subtypes[36]. For ChIP analysis of an IFNα promoter we selected the *IFNα14* gene, whose mRNA was highly inducible in THP-1 cells by dsDNA and confirmed to be MNDA-dependent (Supplementary Fig. 7a). DNA stimulated

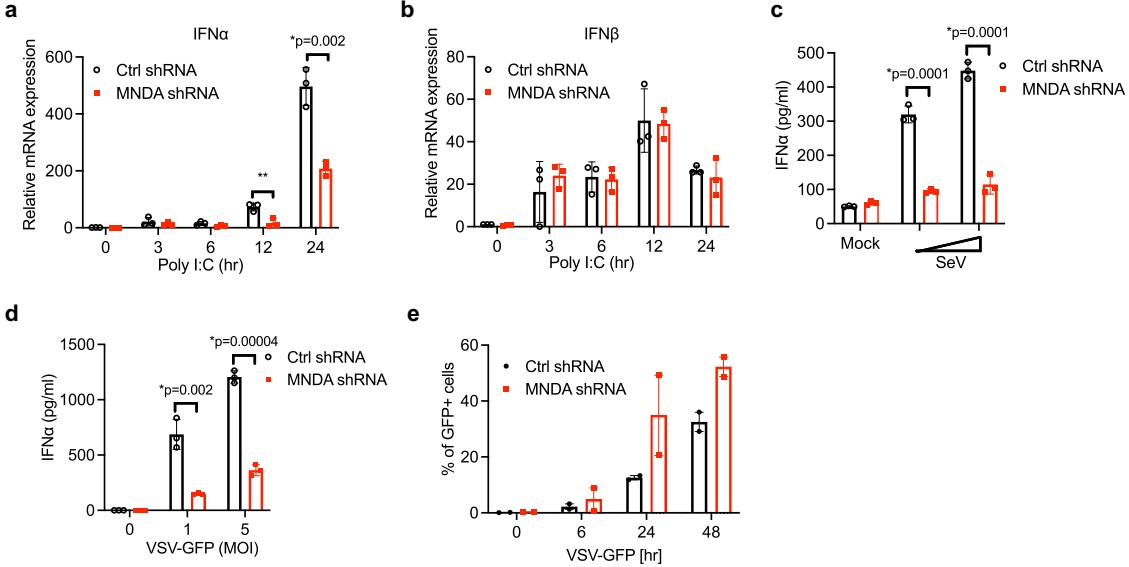

**Fig. 2 MNDA regulates dsRNA- and virus-stimulated IFNα induction in monocytes. a, b** Quantitative PCR analysis of IFNα (**a**) and IFNβ (**b**) mRNA from THP-1 cells expressing control or MNDA shRNA transfected with poly(I:C) (2.5 μg/ml) for the indicated times. **c, d** Release of IFNα protein from THP-1 cells expressing control or MNDA shRNA and infected with Sendai virus (SeV) for 24 h (**c**) or with VSV-GFP at the indicated MOI for 48 h (**d**). **e** Analysis of GFP protein expression by flow cytometry, as a measure of viral replication, in THP-1 cells expressing control or MNDA shRNA infected with VSV-GFP at an MOI of 5 for the indicated times. Data are mean ± SD of triplicate samples and are representative of three independent experiments (**a–d**) or are mean of two experiments (**e**); two tailed unpaired Students t test; *p < 0.05 indicates significance compared to respective groups; ns indicates not significant.

robust Pol II recruitment to this promoter after 8 h stimulation, and this was completely prevented in cells lacking MNDA (Fig. 3c). To explore why IFNα promoter activation was impaired when MNDA was reduced, we next focused on the transcription factor IRF7, since it is the main regulator of IFNα gene promoters[5]. IRF7 is itself an ISG whose expression is upregulated due to PRR-stimulated IFNAR signal transduction[5]. Once IRF7 protein is expressed in cells, similar to IRF3, it is phosphorylated by the PRR-stimulated kinase TBK1, facilitating IRF7 dimerisation, translocation to the nucleus and binding and transactivation of promoters[5]. Figure 3c shows that after DNA stimulation of cells, significant recruitment of IRF7 to the *IFNα14* promoter was observed at 4 h and 8 h. Strikingly, in cells lacking MNDA expression no IRF7 recruitment above baseline levels was seen (Fig. 3c), and this lack of IRF7 recruitment explains impaired formation of the *IFNα14* promoter enhanceosome. As a comparison, we also measured IRF3 recruitment to the *IFNα14* promoter, which was unaffected in cells lacking MNDA (Fig. 3c). Similar results for Pol II, IRF7 and IRF3 recruitment to the IFNa14 promoter were seen for MNDA shRNA cells (Supplementary Fig. 7b–d).

To further explore the relationship between MNDA and IRF7, we examined other IRF7-dependent transcriptional induction events. For the IFNβ promoter, IRF7 controls late induction (Fig. 3a), and like the *IFNα14* promoter, for the *IFNβ* promoter IRF7 but not IRF3 recruitment was impaired in the MNDA shRNA cells (Supplementary Fig. 7e, f). Thus, late but not early recruitment of Pol II to the *IFNβ* promoter was also significantly impaired in MNDA shRNA cells (Supplementary Fig. 7g), since IFNβ induction is initially IRF3-dependent and then becomes more IRF7-dependent once IRF7 expression is upregulated due to IFNAR signalling (Fig. 3a). Further, priming cells with type I IFN in order to increase IRF7 protein expression led to enhanced IFNβ induction after a short (6 h) stimulation with DNA, compared to unprimed cells, and this boost in IFNβ mRNA induction was completely MNDA-dependent (Supplementary Fig. 7h). Type I IFN priming of cells to increase IRF7 protein expression also rendered IFNα mRNA inducible by a 6 h

DNA stimulation, but only in control cells and not in cells with reduced MNDA expression (Supplementary Fig. 7i).

The specificity of MNDA for IRF7-dependent, and not IRF3-dependent transcriptional responses was also demonstrated by examining the mRNA induction of two closely related type III IFN genes, IFNλ-1 and IFNλ-2, whose promoters are IRF3- and IRF7-dependent, respectively[37]. Genetic ablation of MNDA significantly inhibited IRF7-dependent IFNλ-2 mRNA induction (Fig. 3d) while not affecting IRF3-dependent IFNλ-1 mRNA induction (Fig. 3e).

In order to further confirm that MNDA-dependent IFNα induction was via an effect on IRF7, we used IRF7 siRNA in WT and $MNDA^{-/-}$ cells to examine whether IRF7-dependent IFNα induction remained in the absence of MNDA. Figure 3f shows that IRF7 siRNA treatment of WT cells effectively supressed IRF7 induction by DNA. IRF7 siRNA also inhibited DNA-stimulated IFNα induction, and to a similar degree to gene ablation of MNDA (Fig. 3g). Compellingly, in $MNDA^{-/-}$ cells, IRF7 siRNA had no further effect on IFNα induction (Fig. 3g).

Together these results demonstrate that MNDA controls IRF7-dependent but not IRF3-dependent transcriptional responses, and as such controls the IRF7-dependent positive feedback loop in the IFNAR system after pathogen detection by PRRs in monocytes.

**MNDA is required for IRF7 mRNA induction in monocytes and dendritic cells.** To understand why MNDA was required for IRF7-dependent transcriptional responses we next examined whether MNDA was necessary for expression of IRF7 itself. Indeed, compared to WT cells, $MNDA^{-/-}$ cells showed impairment of IFN-stimulated IRF7 protein expression (Fig. 4a). Further, dsDNA-, RNA virus- and type I IFN-stimulated IRF7 protein expression were all diminished in cells with reduced MNDA expression, compared to control cells (Supplementary Fig. 8a–c). Of note, for VSV-GFP infection, impaired IRF7 expression correlated with enhanced GFP protein expression, indicating increased VSV replication (Supplementary Fig. 8b). We next examined the mechanism whereby MNDA regulates

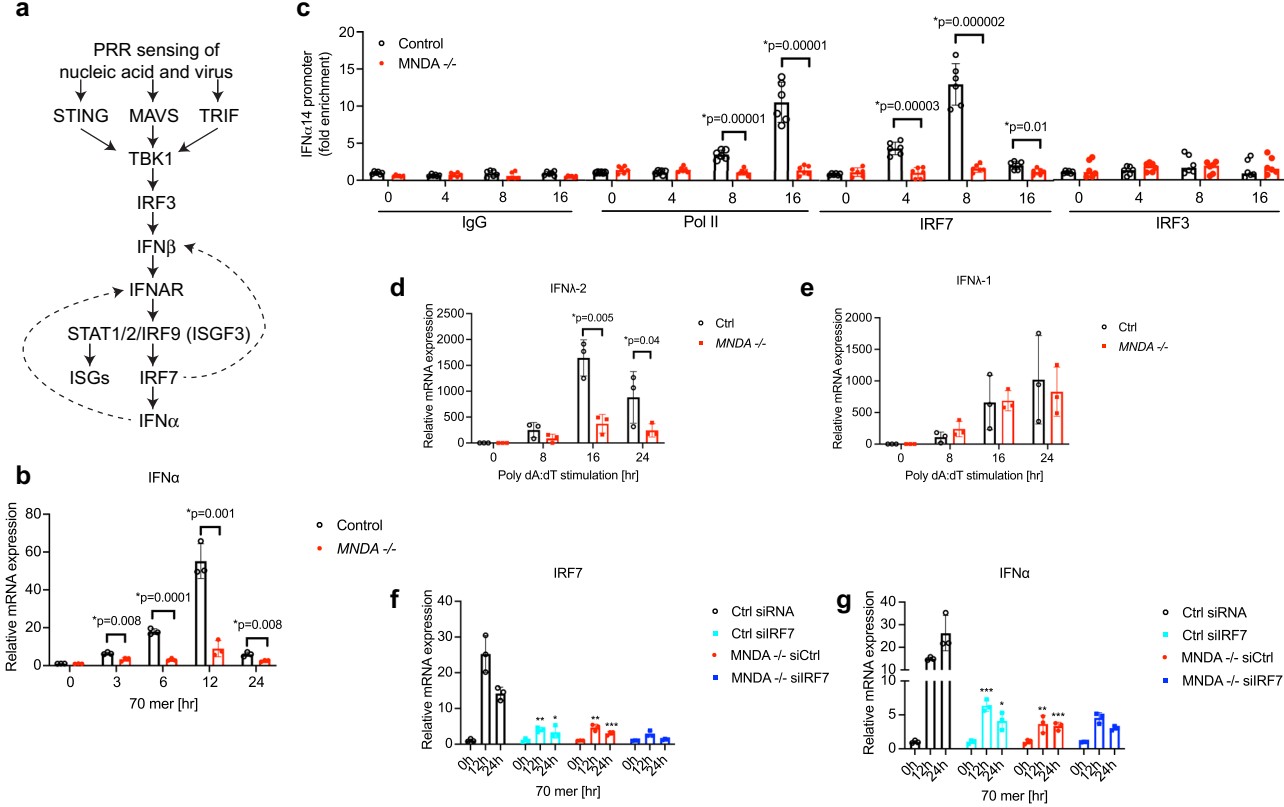

**Fig. 3 MNDA controls IRF7-dependent gene induction. a** Schematic of the PRR-stimulated type I IFN induction cascade. Dashed arrows represent positive feedback loops. **b** Quantitative PCR analysis of IFNα mRNA from three clones of $MNDA^{-/-}$ or control THP-1 cells transfected with dsVACV 70mer (1 μg/ml) for the indicated times. **c** Chromatin immunoprecipitation (ChIP) analysis of the recruitment of RNA Pol II, IRF7 and IRF3 to the IFNα14 promoter in $MNDA^{-/-}$ or control THP-1 transfected with dsVACV 70mer (1 μg/ml) for the indicated times. Sheared chromatin lysates were subjected to ChIP with isotype control (IgG), and anti-RNA Pol II, anti-IRF7 or anti-IRF3 antibodies. Data are presented as mean ± SEM of three independent experiments, each done with technical duplicates (all six data points are shown). **d, e** Three clones of $MNDA^{-/-}$ or control (Ctrl) THP-1 cells were transfected with 2.5 μg/ml poly(dA:dT) for the indicated times. Quantitative PCR analysis of IFNλ-2 (**d**) and IFNλ-1 (**e**) mRNA is shown. **f, g** Quantitative PCR analysis of IRF7 (**f**) and IFNα (**g**) mRNA from $MNDA^{-/-}$ or control (Ctrl) THP-1 cells transfected with 200 pmol of control siRNA or siRNA targeting IRF7 for 24 h, for cells then transfected with dsVACV 70mer (1 μg/ml) for the times indicated. Data are mean ± SD of three clones (**b, d, e**) or mean ± SD of triplicate samples (**f, g**) and are representative of three independent experiments; two tailed unpaired Students $t$ test; $*p < 0.05$ indicates significance compared to respective groups or for (**f**) and (**g**) compared to Ctrl siRNA cells; ns indicates not significant; $p$ values for (**f**) are $*p = 0.001$, $**p = 0.002$, $***p = 0.005$; $p$ values for (**g**) are $*p = 0.008$, $**p = 0.007$, $***p = 0.0002$.

IRF7 protein expression. Studies to date have revealed multiple mechanisms whereby IRF7 protein expression is controlled during cell stimulation, including regulation of protein stability, translation, mRNA stability and promoter induction[5], so we examined which of these, if any, was regulated by MNDA. We first showed that protein stability of IRF7 (or IRF3) was unaltered in cells with reduced MNDA expression (Supplementary Fig. 8d–f). However, gene deletion of MNDA caused significantly reduced induction of IRF7 mRNA in response to either DNA or IFNα stimulation of cells (Fig. 4b–e). Also, the amount of both dsRNA- and dsDNA-stimulated IRF7 mRNA measured was significantly reduced when MNDA expression was reduced (Supplementary Fig. 8g, h), which was also the case for IFNα stimulation (Supplementary Fig. 8i). These data suggest that MNDA regulates IRF7 expression by controlling IRF7 mRNA induction. Further, reduced IRF7 mRNA in MNDA shRNA cells was not due to altered IRF7 mRNA stability (Supplementary Fig. 8j). Other ISGs (IFIT1 and IFIT2) and IRFs (IRF1 and IRF3) were not affected by the absence of MNDA (Fig. 4f–i). Importantly, stable expression of Flag-MNDA in $MNDA^{-/-}$ cells rescued the IRF7 phenotype in that IFNα-stimulated IRF7 mRNA induction was fully restored (Fig. 4j) when Flag-MNDA was expressed in the KO cells (Fig. 4k).

To examine whether MNDA was required for IRF7 mRNA induction in cells apart from monocytes, we next examined IRF7 expression in another myeloid cell type. IRF7 has a particularly important role in pDCs, cells which produce IFNα in response to a viral infection or in a dysregulated manner which contributes to autoimmunity[38]. Because of the rarity of pDCs in human blood, and the difficulty to purify them, we used an established pDC cell line, CAL-1[39], which phenotypically and functionally resemble primary pDCs[40,41] and have been shown to express type I IFN-dependent IRF7[42]. Similar to THP-1 monocytes (Supplementary Fig. 2a), MNDA mRNA was detectable and inducible by IFNα in CAL-1 pDCs (Fig. 5a), as was IRF7 mRNA (Fig. 5b). MNDA shRNA effectively supressed MNDA mRNA induction (Fig. 5a), and protein expression (Fig. 5e), and in parallel prevented IFNα-stimulated IRF7 mRNA induction (Fig. 5b) while not affecting IRF3 nor IRF5 mRNA (Fig. 5c, d).

Thus, in two distinct human myeloid cell models, MNDA is required for IRF7 mRNA induction.

**IFNα-dependent Pol II and STAT2 recruitment to the *IRF7* promoter is MNDA-dependent.** We next focused on the early events surrounding IRF7 mRNA induction and promoter activation. We measured IFNα-stimulated IRF7 mRNA induction in

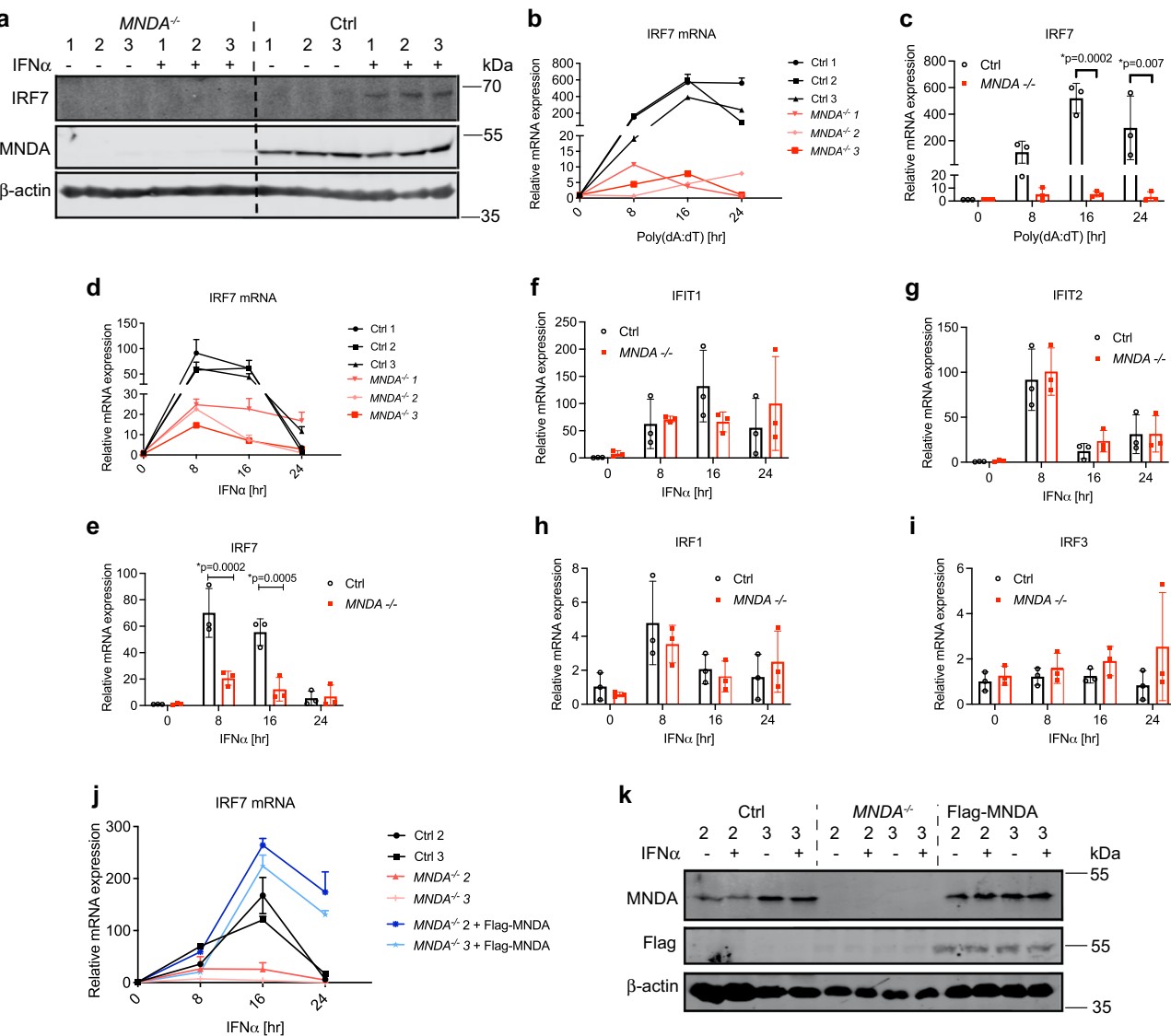

**Fig. 4 Genetic ablation of MNDA impairs IRF7 mRNA induction. a** Immunoblot analysis of IRF7 and MNDA protein expression in three *MNDA*−/− or three control THP-1 clones. **b–e** Three clones of *MNDA*−/− or control (Ctrl) THP-1 cells were transfected with 2.5 μg/ml poly(dA:dT) (**b**, **c**) or treated with 1000 U/ml IFNα (**d**, **e**) or for the indicated times. Quantitative PCR analysis of IRF7 mRNA shown for individual clones (**b**, **d**) or mean of all clones (**c**, **e**). **f–i** Quantitative PCR analysis of mRNA expression of IFIT1 (**f**), IFIT2 (**g**), IRF1 (**h**) or IRF3 (**i**) in *MNDA*−/− or control THP-1 cells. Data shown is mean of three clones. **j**, **k** Flag-MNDA was reconstituted into *MNDA*−/− clone 2 and 3 by lentiviral transduction. Control cells (Ctrl), *MNDA*−/− cells expressing empty lentiviral vector or vector encoding Flag-MNDA were stimulated with IFNα (1000 U/ml) for the indicated times. **j** Quantitative PCR analysis of IRF7 mRNA. **k** Immunoblot analysis of Flag MNDA and MNDA protein after 24 h IFNα stimulation. All data are mean ± SD of triplicate samples and are representative of three independent experiments; two tailed unpaired Students *t* test; *$p < 0.05$ indicates significance compared to respective groups. Immunoblots (**a**, **k**) are representative of three experiments.

the presence of cycloheximide (CHX) which inhibits protein synthesis. This showed that the IRF7 mRNA still measurable when protein synthesis was blocked was still fully MNDA-dependent, since the percentage inhibition of IRF7 mRNA induction in *MNDA*−/− cells was the same in the presence and absence of CHX (Fig. 6a, b), suggesting that MNDA regulates an early promoter-proximal event in IRF7 mRNA induction. Further, in *MNDA*−/− cells stimulated with IFNα for 1 h, Pol II recruitment to the *IRF7* promoter between the region of nucleotides −1000 to +671 was completely impaired, and a ChIP signal was restored in rescued cells stably expressing Flag-MNDA in an anti-Pol II IP (Fig. 6c) compared to an isotype control IP where no ChIP signal was detected (Supplementary Fig. 9a). The absolute requirement for the presence of MNDA to detect Pol II ChIPing to the *IRF7* promoter was also seen in a time course of

IFNα stimulation, for region −500 to −384 (Fig. 6d; isotype control shown in Supplementary Fig. 9b). IFNα stimulation of cells also led to recruitment of Pol II to the *IRF1* promoter, and importantly here there was no difference in Pol II recruitment in *MNDA*−/− cells compared to *MNDA*−/− cells expressing Flag-MNDA (Fig. 6e; isotype control shown in Supplementary Fig. 9c). There was also no difference in the ChIP signal for Pol II between *MNDA*−/− cells and *MNDA*−/− cells expressing Flag-MNDA for a positive control constitutively expressed gene, *EIF4A2* (Fig. 6f), nor for a negative control inactive gene, *SAT2* satellite repeat (Fig. 6g).

To gain further insight into why MNDA was required for Pol II recruitment we considered the role of MNDA in regulation of transcription factors which would be expected to stimulate Pol II recruitment to promoters. Bosso et al. showed that MNDA can

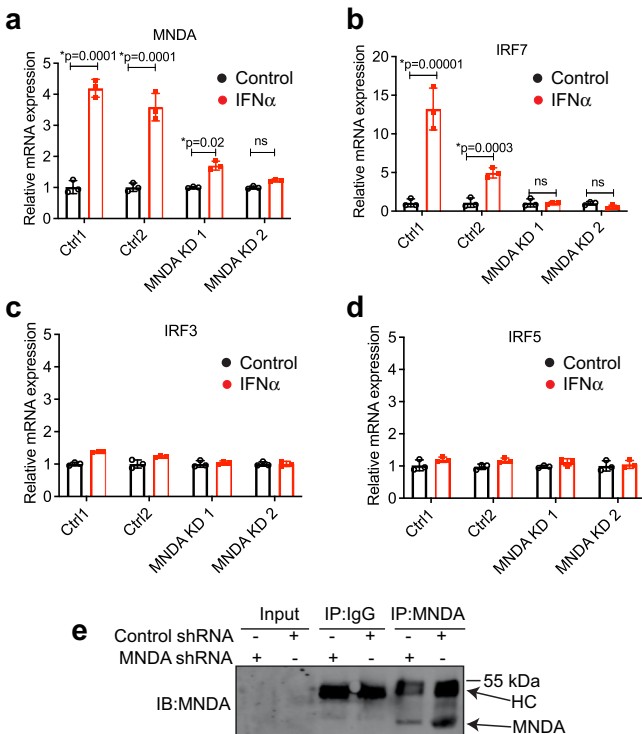

**Fig. 5 MNDA is required for IRF7 induction in pDCs. a–d** Quantitative PCR analysis of MNDA (**a**), IRF7 (**b**), IRF3 (**c**) or IRF5 (**d**) mRNA in CAL-1 cells expressing control or MNDA shRNA and stimulated with IFNα (1000 U/ ml) for 16 h. **e** Anti-MNDA immunoblot (IB) of MNDA immunoprecipitated (IP) from cells treated with control or MNDA shRNA, showing effect of MNDA shRNA on MNDA protein expression. HC, antibody heavy chain. Immunoblot is representative of three experiments. All data are mean ± SD of triplicate samples and are representative of three independent experiments; two tailed unpaired Students $t$ test; *$p < 0.05$ indicates significance compared to respective groups; ns indicates not significant.

interact with Sp1 to restrict HIV-1[23], and since Sp1 has a role in the induction of many human promoters, it was possible that MNDA could regulate the IRF7 promoter through an effect on Sp1. However ChIP analysis of Sp1 binding to the IRF7 promoter showed no role for MNDA in constitutive promoter occupancy by Sp1 (Supplementary Fig. 10a), and neither was Sp1 further recruited to the IRF7 promoter after IFNα stimulation (Supplementary Fig. 10b, c). We then considered other transcription factors known or assumed to regulate human IRF7, and noted that in HeLa cells, STAT2, which forms part of the ISGF3 complex known to regulate ISGs (Fig. 3a), was strongly enriched in its binding to the IRF7 promoter after IFNα treatment of cells[43]. We therefore investigated whether IFNα-dependent STAT2 recruitment to the IRF7 promoter in THP-1 cells was MNDA-dependent, and found that this was indeed the case. Figure 6h shows that the fold enrichment of STAT2 binding to the IRF7 promoter was significantly less in cells lacking MNDA compared to cells expressing Flag-MNDA. Further, in an IFNα time course, significantly less STAT2 was recruited to the IRF7 promoter in the absence of MNDA compared to cells expressing MNDA (Fig. 6i).

We also analysed the presence of positive histone marks on the IRF7 promoter, and consistent with the STAT2 data, showed that for IFNα-stimulated cells, the ChIP signature for Histone H4K5,8,12,16ac (ac-H4), which is associated with active genes, was significantly reduced in cells lacking MNDA compared to Flag-MNDA expressing cells (Fig. 6j). Hence MNDA expression

correlates with and is required for the appearance of active histone marks on the IRF7 promoter.

Thus, MNDA controls IRF7 gene induction by regulating enhanceosome formation on the IRF7 gene promoter, and is required for STAT2, and hence Pol II recruitment to the promoter region.

**MNDA is recruited to the IRF7 promoter after IFN stimulation.** It is possible that PYHIN proteins could directly regulate transcription factor recruitment and gene promoter accessibility by physical interactions with gene promoters, which would be consistent with MNDA being required for recruitment of Pol II to the IRF7 gene promoter, while having no effect on Pol II recruitment to other promoters tested (IRF1, EIF4A2, SAT2). Indeed, ectopic expression of MNDA (but not of other PYHIN proteins) in HEK293T cells, which do not normally express PYHIN proteins, was sufficient to significantly induce an IRF7 promoter-dependent reporter gene, either in the absence or presence of type I IFN priming (Fig. 7a). Further, subcellular fractionation of monocytes confirmed that MNDA is exclusively expressed in the nucleus (Fig. 7b). These data are consistent with a promoter proximal role for MNDA in regulating the IRF7 gene promoter. Therefore, we tested if MNDA itself was associated with the IRF7 promoter, by an anti-Flag ChIP assay to detect Flag-MNDA bound to chromatin in $MNDA^{-/-}$ cells stably expressing Flag-MNDA (Fig. 7d). Remarkably, in the context of cells treated for 1 h with IFNα, Flag-MNDA was found to be associated with the IRF7 promoter (Fig. 7c; isotype control shown in Supplementary Fig. 9a). Interestingly, recruitment of MNDA to the IRF7 promoter was IFNα-dependent, and peaked after 3 h of IFNα stimulation (Fig. 7e; isotype control shown in Supplementary Fig. 9b), while no IFNα-dependent recruitment of MNDA to the IRF1 promoter was observed (Fig. 7f; isotype control shown in Supplementary Fig. 9c)

Together these data reveal that MNDA is a stimulus-dependent regulator of the IRF7 gene promoter in human monocytes and implicate MNDA as a critical regulator of the type I IFN cascade in human myeloid cells.

## Discussion

Although the importance of type I IFNs in health and disease has been firmly established, there is still an urgent need to elucidate the mechanisms whereby type I IFN induction is regulated and controlled, especially in human myeloid cells due to their role in viral sensing and their implication in interferonopthies[1]. Here we show for the first time a requirement for the PYHIN protein MNDA in type I IFN induction in human myeloid cells. Unlike the best characterised human PYHIN protein, IFI16, this was not due to a role for MNDA as a cytosolic PRR for dsDNA. Rather, MNDA was essential for IFNα induction regardless of the PRR stimulating agent. MNDA-dependent IFNα was shown to be due to a requirement for MNDA for induction of the gene encoding IRF7, a transcription factor which occupies a central role in the type I IFN cascade and controls IFNα induction (Fig. 3a)[9]. Consistent with a critical role for MNDA in IRF7 expression, other IRF7-dependent gene expression events were also MNDA-dependent, namely late IFNβ induction and IFNλ-2 induction.

The essential role of IRF7 in the human type I IFN cascade is illustrated by the fact that IRF7 deficiency caused by homozygous loss-of-function variants is linked to life-threatening influenza in children[44], while a loss of function IRF7 variant has also been identified in an adult with severe influenza infection[45]. Furthermore, it was recently shown that there was an enrichment in IRF7 loss-of-function variants in patients with life-threatening COVID-19 pneumonia compared to those with

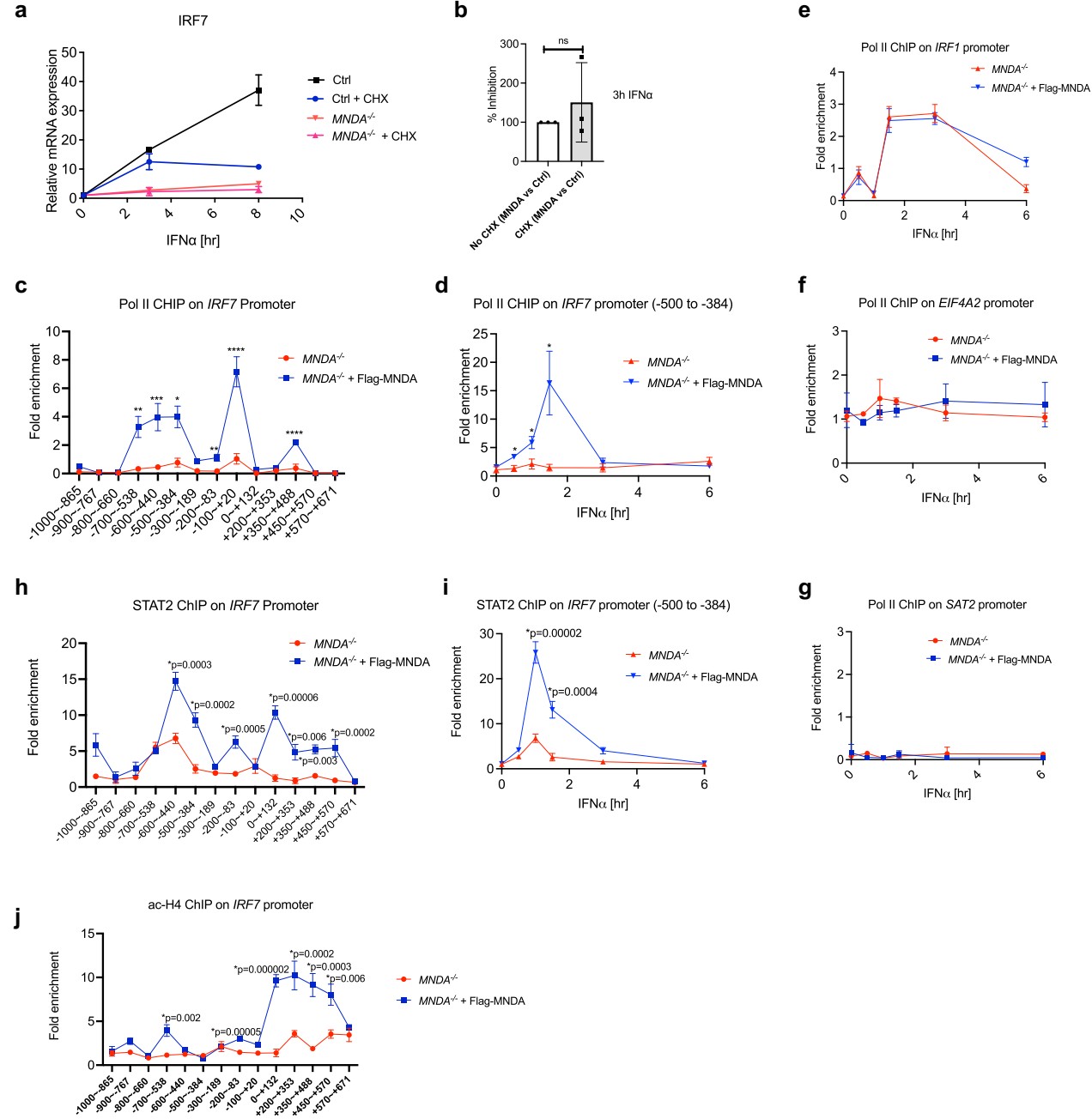

**Fig. 6 IFNα-dependent Pol II and STAT2 recruitment to the IRF7 promoter is MNDA-dependent. a, b** Three THP-1 *MNDA−/−* or control (Ctrl) clones were pre-treated for 1 h with cycloheximide (CHX) as indicated followed by IFNα (1000 U/ml) stimulation for the indicated times. Quantitative PCR analysis of IRF7 mRNA is shown for the average of three cell lines (**a**) and as percentage of inhibition of IRF7 mRNA seen in the *MNDA−/−* cells versus control cells, setting the amount of inhibition in the absence of CHX as 100% (**b**). For (**a**, **b**), data are mean ± SD of triplicate samples and are representative of three independent experiments; two tailed unpaired Students *t* test; significance is defined as *p* < 0.05 between groups; ns indicates not significant (**c**) Chromatin immunoprecipitation (ChIP) of the recruitment of RNA Pol II to the IRF7 promoter between the region of nucleotides −1000 to +671, using primer sets to amplify the specific regions indicated in *MNDA−/−* cells expressing empty lentiviral vector or vector encoding Flag-MNDA. Cells were stimulated with IFNα (1000 U/ml) for 1 hr. **d–g** Cells were treated with IFNα (1000 U/ml) for the indicated times prior to ChIP analysis. ChIP of the recruitment of RNA Pol II to the −500 to −384 region of the IRF7 promoter (**d**), the IRF1 promoter (**e**), the EIF4A2 promoter (**f**) and the SAT2 promoter (**g**) in *MNDA−/−* cells expressing empty lentiviral vector or vector encoding Flag-MNDA. **h** ChIP of the recruitment of STAT2 to the IRF7 promoter between the region of nucleotides −1000 to +671, using primer sets as per (**c**). Cells were stimulated with IFNα (1000 U/ml) for 1 hr. **i** ChIP of the recruitment of STAT2 to the −500 to −384 region of the IRF7 promoter. Cells were treated with IFNα (1000 U/ml) for the indicated times. **j** ChIP of the recruitment of Histone H4K5,8,12,16ac (ac-H4) to the IRF7 promoter between the region of nucleotides −1000 to +671, using primer sets as per (**c**). Cells were stimulated with IFNα (1000 U/ml) for 1 hr. For (**c–j**), data are presented as mean ± SEM of three independent experiments, each done with technical duplicates; two tailed unpaired Students *t* test; *p* < 0.05 indicates significance compared to respective groups; *p* values for (**c**) are *p* = 0.02, **p* = 0.003, ****p* = 0.004, *****p* = 0.0002; for (**d**) are *p* = 0.02. ChIP antibody isotype controls for (**c–e**, **h–j**) are shown in Supplementary Fig. 9.

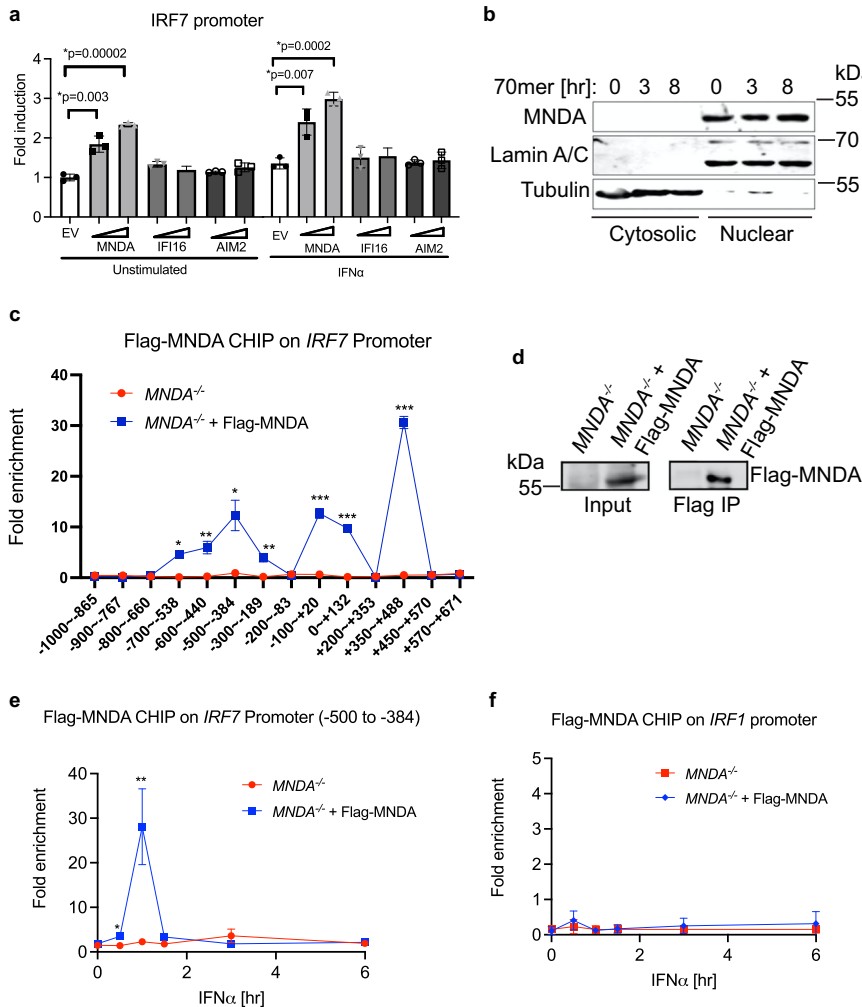

**Fig. 7 MNDA is recruited to the IRF7 promoter after IFN stimulation. a** Assessment of the effect of expression of PYHIN proteins on IRF7 promoter induction by reporter gene assay. HEK293T cells were transfected with pCMV-HA empty vector (EV) or the same vector expressing MNDA, IFI16 or AIM2 for 24 h prior to stimulation with IFNα (1000 U/ml) for a further 24 h. Data are mean ± SD of triplicate samples and are representative of three independent experiments; two tailed unpaired Students t test; *p < 0.05 indicates significance compared to respective groups. **b** Analysis of subcellular expression of MNDA protein by immunoblot of lysates from THP-1 cells transfected with dsVACV 70mer (1 μg/ml) for the indicated times. Cytoplasmic and nuclear fractions generated were immunoblotted for the indicated proteins. Representative of three experiments. **c** ChIP of the recruitment of Flag-MNDA to the IRF7 promoter between the region of nucleotides −1000 to +671, using primer sets to amplify the specific regions indicated, in $MNDA^{-/-}$ cells expressing empty lentiviral vector or vector encoding Flag-MNDA. Cells were stimulated with IFNα (1000 U/ml) for 1 hr. **d** Anti-Flag immunoprecipitation (IP) and immunoblot analysis of Flag-MNDA protein from (**c**). **e, f** ChIP of the recruitment of Flag-MNDA to the −500 to −384 region of the IRF7 promoter (**e**) or the IRF1 promoter (**f**) in $MNDA^{-/-}$ cells expressing empty lentiviral vector or vector encoding Flag-MNDA. Cells were treated with IFNα (1000 U/ml) for the indicated times. For (**c, e, f**), data are presented as mean ± SEM of three independent experiments, each done with technical duplicates; two tailed unpaired Students t test; *p < 0.05 indicates significance compared to respective groups; p values for (**c**) are *p = 0.001, **p = 0.003, ***p = 0.00001; p values for (**e**) are *p = 0.04, **p = 0.01. ChIP antibody isotype controls for (**c, e, f**) are shown in Supplementary Fig. 9.

asymptomatic or mild infections, implicating IRF7 in anti-SARS-CoV-2 responses[46].

Although these studies illustrate the critical role of IRF7 in human viral disease, compared to IRF3, especially in human, much less is known about how IRF7 is regulated. Human IRF7 is a classic ISG, induced by the IFNAR-dependent transcription complex ISGF3[8]. However here we show a further layer of complexity in IRF7 gene induction in myeloid cells, since other ISGF3-dependent ISGs such as IFITs, IRF1 and ISG15 did not require MNDA. A rationale for the requirement of MNDA in IRF7 promoter induction was revealed by ChIP assays of the IRF7 promoter. Generation of both MNDA shRNA-expressing cells, or of $MNDA^{-/-}$ cells revealed that MNDA was required for stimulus-induced recruitment of Pol II to the IRF7 gene promoter. This could be explained by the fact that MNDA was

also required for IFNα-stimulated recruitment of STAT2 to the IRF7 promoter, since STAT2 is known to regulated the human IRF7 promoter, presumably as part of the ISGF3 complex[43]. After IFNα stimulation, we detected STAT2 binding to the IRF7 promoter at regions known to contain the IFN-stimulatory response element (ISRE) that STAT2 binds to, namely upstream of the TSS near position −250, and downstream of the TSS in the 5'UTR near position +250[47,48]. We also detected a strong peak of binding further upstream between −500 and −384. In all cases, STAT2 binding was partially or fully dependent on the presence of MNDA. In correlation with the requirement of MNDA for STAT2 recruitment, MNDA-expressing cells showed a significant enrichment of activating histone marks (H4-ac) compared to MNDA-deficient cells in response to IFNα stimulation.

How exactly MNDA regulates STAT2 recruitment and promoter accessibility for IRF7 is unclear but interestingly we found that MNDA itself was actually recruited to the *IRF7* promoter. Reconstitution of *MNDA*[−/−] cells with Flag-MNDA protein provided a system to examine recruitment of MNDA to the *IRF7* promoter. We selected region -1000 to +671 which encompasses promoter elements known to be required for virus- and IFN-stimulated *IRF7* promoter induction[47,48]. The ChIP signal for MNDA showed several peaks within this region, and when we looked at one region in particular, −500 to −384, we observed IFNα stimulation-dependent recruitment of MNDA to the promoter. Although other PYHINs have previously been implicated in regulation of transcription[49,50], this to our knowledge is the first demonstration of a positive role of a PYHIN in IRF promoter induction. These results implicate MNDA as a direct stimulus-dependent regulatory factor for IRF7 gene induction in human myeloid cells. Since positive feedback of IRF7-dependent IFNα is a major source of type I IFN during antiviral immune responses, MNDA thus occupies a central role in the myeloid cell contribution to such responses. Consistent with this we saw enhanced RNA virus replication in infected monocytes where MNDA expression was supressed.

In contrast to the role of human MNDA in positive regulation of IRF7 gene induction, a recent study of mouse PHYINs found an opposite role in IRF7 regulation[51]. There, siRNA knockdown of mouse PYHIN IFI204 inhibited RNA virus-stimulated IFNα gene induction, while IFI204 was shown to physically interact with IRF7 protein in the nucleus, and promote IRF7 nuclear retention, yet inhibit IRF7 binding to gene promoters. The authors also showed that human PYHINs, including MNDA, when overexpressed could co-immunoprecipitate with IRF7 and when expressed in mouse fibroblasts lacking both IRF3 and IRF7, MNDA inhibited type I IFN gene induction stimulated by overexpressed IRF7. In contrast, here we showed a positive role for MNDA in IRF7-dependent IFNα induction using multiple approaches including siRNA, shRNA, CRISPR/Cas9 and rescue experiments and ChIP. Therefore, the role of PYHIN proteins in IRF7 regulation is likely to be complex and there may be fundamental differences in this regard in mouse versus human. In contrast to the five human PYHINs, the mouse locus encodes 13 PYHINs, with no direct MNDA ortholog (even though one of the mouse genes is named Mnda)[52,53].

Our data raises several important questions relating to the ability of MNDA to regulate transcription. How exactly MNDA, which resides in the nucleus in unstimulated cells, is mobilised to the *IRF7* promoter in response to IFNAR signalling is currently unclear. Also, it is unclear whether MNDA physically associates directly with the IRF7 promoter, or is part of a larger protein complex that regulates promoter accessibility. Further, it will be interesting to investigate the exact contribution of MNDA recruitment to the *IRF7* promoter in facilitating STAT2 recruitment and activating histone marks. Interestingly, IFI16 has also been recently shown to be recruited in a stimulus-dependent manner to the RIG-I gene promoter in influenza A virus-infected cells[54].

MNDA may regulate *IRF7* promoter induction by engaging with other proteins involved in enhanceosome formation at promoters, since it has been shown to associate with the transcription factors YY1[55] and Sp1[23], as well as with methyltransferases that can regulate gene expression[56] and with other proteins that indirectly affect transcription factor activity[25]. An important overall question for PYHIN regulation of transcription is the need to understand how the specificity of a PYHIN for a particular gene or sets of genes is defined. Why MNDA regulates the *IRF7* promoter but not other ISGF3-dependent promoters in myeloid cells, why IFI16 is recruited to the RIG-I promoter[54], and why PYHIN1 is required for IL-6 and TNF but not IL-8 induction in epithelial cells is not yet clear[26]. Although PYHINs can bind to dsDNA via their HIN domain, the crystal structures of the AIM2 and IFI16 HIN domains bound to dsDNA showed that these proteins engage with the dsDNA phosphate backbone, and not with specific DNA residues, so although PYHINs can bind dsDNA, promoter specificity is unlikely to be defined in the HIN domain.

Given the apparent specificity of MNDA for IRF7 induction, and the cell type-specific expression of MNDA, this PYHIN could be a key target either to boost type I IFN during a viral infection, or to inhibit dysregulated type I IFN during autoimmunity[57]. For example, in SLE, a systemic autoimmune disease, an elevated peripheral IFN signature consistent with induction of type I IFN characterises the disease[1], and in SLE patients, blood monocytes show an especially prominent type I IFN response as do pDCs[1], two cell types in which we have shown a role for MNDA in controlling IFNα production. In conclusion, we have shown that MNDA is a positive regulator of the type I IFN cascade in human myeloid cells, due to a promoter-proximal role in IFN-stimulated IRF7 gene induction. Overall, the study emphasises the emerging role of human PYHINs as critical regulators of cytokine and IFN induction in response to pathogens.

## Methods

**Cell culture**. THP-1, HEK293 and HEK293T cells were purchased from the European Collection of Cell Cultures. HEK-Blue IFN-α/β reporter cells were purchased from InvivoGen. The human pDC cell line CAL-1 were a gift from Dr T. Maeda, Department of Island and Community Medicine, Nagasaki University, Japan[39]. HEK293T cells were grown in DMEM, supplemented with 10% FCS (v/v) and 10 µg/ml penicillin-streptomycin. THP-1 and CAL-1 cells were grown in RPMI medium, supplemented with 10% FCS (v/v) and 10 µg/ml penicillin-streptomycin. HEK-Blue IFN-α/β reporter cells were grown in DMEM medium with 10% FBS containing selection antibiotics Blasticidin (30 µg/ml), Zeocin (100 µg/ml) and Normocin (100 µg/ml). All cells were kept at 37 °C with 5% $CO_2$. For THP-1 differentiation, cells were treated with 100 nM of PMA for 24 h prior to stimulation. Cells were previously confirmed as mycoplasma-free.

**Primary human cells**. Ethical approval was obtained from the TCD School of Biochemistry and Immunology Research Ethics Committee for experiments involving PBMCs. Human peripheral blood mononuclear cells (PBMCs) from anonymous healthy donors were obtained by informed consent from buffy coats of blood packs from the Irish Blood Transfusion Service, using Lymphoprep (Axis-Shield) gradient centrifugation. For cell sorting to analyse constituent cells, PBMCs were stained with anti-CD14 APC, anti-CD19 PE, anti CD3 PE-Cy5.5 and anti-CD56 PE-Cy7 (eBiosciences). Cells were fixed and permeabilised using the FoxP3 staining buffer set (eBiosciences) and were stained with anti-MNDA FITC (Cell Signalling). Labelled cells were analysed on a FACSCanto II flow cytometer (BD Biosciences) and were evaluated with FlowJo software (TreeStar). Monocytes were isolated from PBMCs by positive selection using CD14 beads (Miltenyi Biotech), following the manufacturer's instructions. Macrophages were generated by growing the monocytes in the presence of 50 ng/ml granulocyte macrophage colony stimulating factor (GM-CSF, Sigma–Aldrich) or 20 ng/ml macrophage colony stimulating factor (M-CSF, Sigma–Aldrich), for 7 days, changing the media every 2–3 days.

**Cell treatments with stimulants and viruses**. The vaccinia virus (VACV) 70 bp dsDNA oligonucleotide (dsVACV 70mer) was synthesised by MWG Biotech and has been described previously[10]. For cell stimulations, cells were transfected with dsVACV 70mer (1 µg/ml) or low molecular weight poly(I:C) (Invivogen, 2.5 µg/ml) using Lipofectamine 2000 (1 ul/ml, Invitrogen), or stimulated with 1000 U/ml human rIFNα (PBL Assay Science). Sendai virus (SeV) Cantell strain was from ATCC. VSV-GFP refers to vesicular stomatitis virus Indiana serotype, attenuated due to a deletion of methionine 51 in the gene encoding the matrix protein and containing a transgene encoding enhanced green fluorescent protein, and has been described[58]. For cycloheximide (CHX) treatment, 25 µg/ml CHX was added either 1 h prior to IFNα stimulation (for mRNA induction experiment) or 16 h after IFNα simulation (for protein stability experiment).

**VSV replication assay**. Cells were infected for 48 hr with VSV-GFP. GFP-positive cells were analysed on a FACSCanto II flow cytometer (BD Biosciences) and were evaluated with FlowJo software (TreeStar).

**Cell fractionation**. Cells were washed in PBS and lysed in cytosolic lysis buffer (10 mM NaCl, 3 mM MgCl$_2$, 1 mM EDTA, 1% Triton X-100, 1 mM Na$_3$VO$_4$, 1 mM PMSF, 1% aprotinin, 10 mM Tris/HCl, pH 7.4) on ice for 30 min. The lysate was centrifuged at 1000 g for 30 min and the supernatant containing the cytosolic proteins was collected. The pellet was washed twice in cytosolic lysis buffer and then resuspended in nuclear lysis buffer (1.5 mM MgCl$_2$, 0.42 M NaCl, 0.2 mM EDTA, 25% glycerol, 1 mM DTT, 1 mM Na$_3$VO$_4$, 1 mM PMSF, 1% aprotinin, 20 mM HEPES, pH 7.9) and placed on ice for 30 min. The lysates were centrifuged at 16,000 g for 10 min and the supernatant containing the nuclear proteins was collected.

**Immunoblotting**. Protein extracts were prepared by lysing cells with sample buffer (6% SDS, 30% glycerol, 0.3% bromophenol blue, 150 mM DTT, 187.5 mM Tris, pH 6.8), boiled and subjected to SDS-PAGE and immunoblotting. Primary antibodies used were anti-MNDA (3C1 at 1/2000), anti-phospho-IRF3 (4D4G at 1/1000), anti- IRF7 (4920 S at 1/1000), anti-phospho-STAT1 (Y701 at 1/1000) and anti-STAT1 (9171 S at 1/1000) from Cell Signalling; anti-IRF3 (18781 at 1/1000) from Immuno-Biological Laboratories; anti-IFI16 (1G7 at 1/1000), anti-GFP (B-2 at 1/1000) and anti-Lamin A/C (636 at 1/1000) from Santa Cruz Biotechnology; anti-AIM2 (3B10 at 1/1000) from Adipogen; anti-β-actin (AC-74 at 1/100,000) from Sigma–Aldrich; anti-tubulin (DM1A at 1/5000) from Millipore; anti-PYHIN1 (used at 1/1000; a gift from Jin-Fong Lee (University of Texas, USA)[59]; anti-IFIT3 (used at 1/2000; a gift from Andreas Pichlmair (School of Medicine, Technical University of Munich, Germany); anti-Flag (M2 at 1/2000) from Sigma–Aldrich. The next day, membranes were incubated with secondary Abs and blots were visualised using the Odyssey imaging system (LI-COR Biosciences). ImageJ software version 1.42 was used for image acquisition and densitometric analysis of immunoblots. The rectangular selection in the programme tools were used to specify the area of intensity measured for each band. The intensity of each band was then normalised with the intensity of corresponding protein loading controls. The uncropped version of all immunoblots are shown in the source data file.

**RNA-mediated interference**. THP-1 cells stably expressing shRNA targeting MNDA and IFI16, or a scrambled control shRNA, or CAL-1 cells stably expressing shRNA targeting MNDA were generated using the lentiviral pLKO.1 vector (Sigma–Aldrich). The IFI16- and MNDA- silencing sequences targeting coding sequences of both genes were from the MISSION TRC-Hs 1.0 (Human) and were 5'-GCAAATTATGTTTGCCGCAAT-3' (TRC identifier: TRCN0000019079) and 5'-CCTTGTTAACAATCTTCGAAA-3' (TRC identifier: TRCN0000020003), respectively. Lentiviral particles were produced in HEK293T cells transfected with 4 µg of shRNA along with 3 µg of pSPAX and 1 µg of pMD2 for 48 h. Viral supernatant was collected, filtered, and then added to target THP-1 cells. THP-1 cells with shRNA knockdown were selected by puromycin (150 µg/ml) 48 h later. Cells containing the pLKO.1 lentiviral shRNA expression vectors were then cultured in puromycin (1 µg/ml).

To demonstrate knockdown of MNDA protein expression in CAL-1 cells following shRNA treatment, $5 \times 10^7$ CAL-1 cells stably expressing shRNA targeting MNDA or scrambled control shRNA were lysed with 2 ml of IP lysis buffer containing 50 mM Hepes, 100 mM NaCl, 1 mM EDTA, 10% glycerol and 1% NP40, and the inhibitors 1% aprotinin, 1 mM PMSF, and 1 mM sodium orthovanadate. The cell lysates were centrifuged at 14,000 g for 10 min at 4 °C. Endogenous MNDA from CAL-1 cells was immunoprecipitated from the cell lysates using protein A/G beads coupled with anti-MNDA or isotype control Ab overnight at 4 °C and samples immunoblotted for MNDA.

For transient siRNA of THP-1 cells or primary human monocytes, cells were transfected with siRNA using the Neon electroporator (ThermoScientific). For primary monocytes, SMARTpool: ON-TARGET siRNA against MNDA (Dharmacon) was used. Primary monocytes were electroporated with 200 pmol of siRNA per $2 \times 10^6$ cells, using electroporation setting 9 and then incubated for 72 h before stimulation. For THP-1 cells, siRNA against MNDA was from Qiagen (Targeting sequence 5'-CCTTGTTAACAATCTTCGAAA-3'), or SMARTpool: ON-TARGET siRNA against IRF7 was from Dharmacon. THP-1 cells were electroporated with 50 pmol and 100 pmol of siRNA per $2 \times 10^6$ cells, using electroporation setting 15 and then incubated for 24 h before stimulation.

**RNA analysis by quantitative RT-PCR**. Total RNA was extracted from cells using the High Pure RNA Isolation Kit (Roche) and reverse transcribed with random hexamers (IDT) using Moloney murine leukemia virus reverse transcriptase (Promega) according to the manufacturer's instructions. mRNA was quantified with SYBR Green using primer pairs targeting MNDA, IRF7, IFIT1, IFIT2, IRF1, IRF3, IRF5 and β-actin (Primer sequences are listed in Supplementary Table 1). Relative mRNA expression was calculated using the comparative C$_T$ method, normalising the gene of interest to the housekeeping gene β-actin, analysing the data as fold induction compared to that of the control sample.

**ELISA**. Quantification of secreted human IFNα from cell supernatants was measured by Human IFN-α pan ELISA BASIC kit (Mabtech) following the manufacturer's instructions.

**IFN-α/β bioassay**. Supernatants and IFNα standard were diluted in the test medium (DMEM, 10% FCS, 50 units/ml penicillin, 50 µg/ml streptomycin, 100 µg/ml normocin). HEK-Blue IFNα/β reporter cells ($2.8 \times 10^5$ cells/ml) were seeded in 96-well plates containing supernatant, standard and blank. Following 24 h incubation at 37 °C, secreted SEAP was detected by QUANTI-Blue (Invivogen), and absorbance was measured at 620 nm.

**Generation of MNDA$^{-/-}$ THP-1 cells**. THP-1 cells stably expressing Cas9[34,35] were seeded at $5 \times 10^5$ cells/ml on 6-well plates. 180 pmol of MNDA single guide RNA (sgRNA) 1 or 2 (gRNA1: 5'-ATTTAGGACTAACTACA-3'; gRNA2: 5'-AGCTATAACATCAGAAATGG-3') were transfected into THP-1 cells by using Neon Electroporation Transfection at 1400 V for $3 \times 10$ msec pulses according to the manufacturer's protocol. Control cells were transfected with sgRNA targeting GFP (5'-AGCTGGACGGCGACGTAAA-3'). At 72 h after transfection, single cell sorting of living cells was done to obtain a cell density of 1 cell per well on a 96-well plate by Flow Cytometer. Immunoblotting was performed to select clones lacking MNDA expression, which were then confirmed by NGS to have disrupted MNDA alleles.

**MNDA-flag reconstitution of MNDA$^{-/-}$ cells**. Retroviral transduction was used to express Flag MNDA in THP-1 MNDA$^{-/-}$ cell clone 2 and 3. C-terminal Flag-tagged MNDA was cloned from human genomic cDNA into retroviral expression vector pDI[60] using forward primer (GGACTAGTCCACCATGGTGAATGAATAC AAG) and reverse primer (GACCAATGAATGTTAATGCGGCCGCCGACTACA AGGACGACGACGACAAGTGAACGCGTGGG). Retroviral particles were produced in HEK293T cells: cells were seeded at $2 \times 10^5$ cells/ml in 10-cm dishes and transfected 24 hr later with 3 µg pDI empty vector control or pDI MNDA-Flag and transfected along with VSV G protein (pMD-G) and Gag/Pol protein (pCMVR8.91) into HEK 293 T cells, according to the method by Chinnakannan et al. (2013)[61]. The resulting viruses were introduced into MNDA$^{-/-}$ clone 2 and 3 by spinoculation. 48 h later puromycin at a concentration of 5 µg/ml was added to the MNDA$^{-/-}$ clone 2 and 3 to select the MNDA-Flag expressing cells.

**Chromatin immunoprecipitation (ChIP) analysis**. $5 \times 10^7$ THP-1 cells were seeded in a 10 cm dish, then stimulated for the indicated times. After stimulation, cells were centrifuged and fixed with 1% (v/v) formaldehyde for 10 mins with gentle shaking. The formaldehyde was quenched with 0.125 M glycine for 5 min. Cells were washed 3 times with PBS and lysed with 500 µl of ChIP lysis buffer (150 mM NaCl, 5 mM EDTA, 0.5% NP-40 and 1% Triton X-100, 1 mM Na$_3$VO$_4$, 1 mM PMSF, 1% aprotinin, 50 mM Tris-HCl, pH 7.5). Cells were sonicated using a Bioruptor® Pico sonicator for 15 cycles of 30 s pulses, with 30 s rest between each cycle. Sheared chromatin was cleared by centrifugation and 50 µl of samples were taken out for input. Samples (100 µl) were then incubated overnight at 4 °C with 2 µg of the anti-Pol II (N-20 X; Santa Cruz), anti-IRF7 (H-246 X; Santa Cruz), anti-IRF3 (FL-425 X; Santa Cruz), anti-Flag (M2; Sigma–Aldrich), anti-Sp1 (rabbit polyclonal, Abcam), anti-STAT2 (B-3, Santa Cruz), anti-Histone H4K5,8,12,16ac (rabbit polyclonal, Merck) or an isotype control (IgG), while rotating. The following day, protein A–Sepharose beads were blocked for 45 min with 100 µg salmon sperm DNA (Invitrogen) and 0.5 mg BSA per 1 ml beads (50% slurry in ChIP buffer), then washed once in ChIP buffer. Blocked beads were incubated with cleared chromatin immunocomplexes for 1 h at 4 °C with rotation. Beads were then washed five times with ChIP buffer and eluted with 250 µl of Elution buffer (1% SDS and 0.1 M NaHCO$_3$) at 65 °C for 15 mins. The eluate and input were centrifuged and reverse the cross-links by adding 0.4 M NaCl for overnight shaking at 55 °C. The following day, 1 µl proteinase K (20 µg/ml; Qiagen) was added in samples and incubated for 40 min at 55 °C while shaking. Samples were then subjected to PCR purification kit (Qiagen) to extract DNA. Purified input and IP DNA were analysed by qRT-PCR using primers specific for the promoter region of target genes (Supplementary Table 2). Results were normalised to input and are presented as fold enrichment relative to the untreated control.

**Reporter gene assay for IRF7 promoter induction**. MNDA, IFI16 and AIM2 were cloned into the expression vector pCMV-HA (Clontech). The firefly luciferase reporter pGL3-IRF7 promoter was a kind gift from Fanxiu Zhu (Florida State University, Tallahassee, USA). Luciferase reporter gene assays were performed in HEK293 cells seeded in 96-well plates and transfected with polyethyleneimine (Sigma–Aldrich). Firefly reporter plasmid (60 ng), 20 ng GL3-Renilla control plasmid and 50 ng and 100 ng expression vector or empty vector control were used per well. Cells were lysed in Passive Lysis Buffer (Promega), cell lysates assayed in a luminometer, and firefly luciferase activity was normalised to Renilla luciferase activity. Data are expressed as mean fold induction with standard deviations (SD) relative to control levels for an individual experiment performed in triplicate.

**Statistical analysis**. Data were analysed using two tailed unpaired Student's $t$ test.

**Reporting summary**. Further information on research design is available in the Nature Research Reporting Summary linked to this article.

## Data availability

All data generated or analysed during this study are included within this article (and its supplementary information files). Source data are provided with this paper.

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

## Acknowledgements

This work was funded by the Irish Higher Education Authority (HEA) under the Programme for Research in Third Level Institutes (PRTLI) co-funded by the Irish Government and the European Union, and grants from Science Foundation Ireland (11/PI/1056 & 16/IA/4376) and the National Institutes of Health (AI093752), and by a Biotechnology and Biological Sciences Research Council-Science Foundation Ireland joint award (BBSRC-SFI, BB/P020194/1 to A.G.B.).

## Author contributions

Conceptualisation: D.C., L.G., G.B., L.U. and A.G.B. Methodology: L.G., D.C. and G.B. Investigation: L.G., D.C., G.B. and A.P.B. Resources: S.C. and K.A.F. Writing – original draft: A.G.B., D.C. and L.G. Supervision: A.G.B. Funding acquisition: A.G.B.

## Competing interests

The authors declare no competing interests.
