## [Peer Review File · Nature Communications]

Myeloid cell nuclear differentiation antigen controls the pathogen-stimulated type I interferon cascade in human monocytes by transcriptional regulation of IRF7REVIEWER COMMENTS

Reviewer #1 (Remarks to the Author):

In the study entitled “MND A controls the pathogen-stimulated type I interferon cascade in human monocytes by transcriptional regulation of IRF7” by Casserly et al., the authors show that the PYHIN protein MND A is required for IFN type I induction in monocytes. Unlike other PYHINs, this was not due to a pathogen sensing role, but rather MND A regulated expression of IRF7, a transcription factor essential for type I IFN induction. Mechanistically, MND A was required for recruitment of RNA polymerase II to the IRF7 gene promoter, and in fact MND A was itself recruited to the IRF7 promoter after type I IFN stimulation. Overall, these data provide evidence that MND A is a critical regulator of the type I IFN cascade in human myeloid cells. Although this conclusion appears novel and interesting, unfortunately the study suffers from several flaws that should be addressed in order to strength the data and make it suitable for publication.

General points:

1. MND A, expressed in myelomonocytic and B lymphoid hematopoietic cells, associates with chromatin, but does not bind specific DNA sequences (Xie J. et al., J Cell Biochem., 1998). MND A participates in a ternary complex with the transcription factor YY1 target DNA element. According to this may the authors investigate if in their system MND A binds DNA even at low affinity? The results achieved could shed some light on the mechanisms MND A relies on to recruit RNA polymerase II on the IRF7 promoter.
2. Does MND A directly interact with RNA polymerase II and, in positive case, may the authors provide information about the domains involved in such an interaction?
3. It has been recently demonstrated (Bosso et al., PLOS Pathogens, 2020) that MND A interacts with some transcription factors, including Sp1, in order to restrict HIV replication in macrophages. On the light of these results, the authors should investigate by Electrophoretic Mobility Shift Assay along with antibody supershift and Chromatin Immunoprecipitation if other transcription factors are recruited or bound on the IRF7 promoter where they might interact with MND A after type I IFN stimulation. Otherwise, as it is now, a potential reader may get the impression that MND A is the only transcription factor driving the activity of the IRF7 gene promoter. This cannot be true when dealing with such a complex promoter as that of IRF7.

Specific points:

1. Introduction, line 68. The reference showing that IFI16 restricts the growth of HPV (Lo Cigno et al. J.Virol, 2015) in addition to Herpesviruses and HIV is missing and should be also mentioned in the text.
2. Figure 1C, line 113-114. “Since MND A protein expression was higher in monocytes than macrophages in contrast to IFI16 where the opposite was the case...” This comment does not correspond to the data presented in the Immunoblot showing that both unprimed or PMA-primed THP-1 express comparable levels of the IFI16 protein. The sentence should be modified. Moreover, the authors should explain why the IFN-beta mRNA levels of control shRNA cells upon 70mer stimulation (panel E, >20 fold) are much lower when compared to those reported for the same treatment in the same cells in the kinetics shown in panel D, 100-200 fold. They should be similar. In addition, to substantiate their data, the authors should also performe kinetic experiments with IFI16-depleted cells as performed with MND A-depleted cells.
3. Figure 1E. Kinetics experiments for IFI16 shRNA should be included as in panel D.
4. Figure 1I-K. To provide compelling evidence about IFN-alpha regulation, it is mandatory to quantify the protein in the culture supernatants by ELISA or similar assays.
5. Figure 3B-C. As shown in panel B, the IFNalpha14 mRNA is highly inducible at 12 h in THP- 1 cells upon stimulation with 70mer. By contrast, robust Pol II recruitment is detectable 16 h after stimulation suggesting that IFNalpha14 mRNA expression precedes its transcription by Pol II. This discrepancy should be explained.

6. Figure 3H-J. “Interestingly, dsDNA-, RNA virus- and type I IFN-stimulated IRF7 protein expression were all diminished in cells with reduced MNDA expression, compared to control cells”. This conclusion does not seem to be supported by the included immunoblot. In THP-1 cells expressing control or MNDA shRNA transfected with dsVACV or stimulated with IFN- α , IRF7 expression is not significantly decreased at 24 h after stimulation going against the conclusion drawn by the authors. This discrepancy must be justified.

7. Figure 3M. The indicated times considered for the analysis should be the same as in figures 3K and 3L.

8. Figure 4. The authors should perform the same experiments of supplementary figures 2E and 2F in Cal-1 pDCs shRNA-treated cells to prove the reduction of MNDA’s expression. To this purpose, an Immunoblot analysis of MNDA protein expression in CAL-1 pDCs cells stably expressing control or MNDA shRNA, and a quantitative PCR analysis of MNDA mRNA from CAL-1 pDCs cells stably expressing control or MNDA shRNA should be performed.

9. Figure 5. To confirm the removal of MNDA genetic locus with the CRISPR/Cas9 technology, the authors should perform a TIDE analysis.

10. Figures 6-7. In this set of experiments the authors demonstrate that IFN- α -dependent Pol II recruitment to the IRF7 promoter is MNDA-dependent. Although they show that MNDA is constitutively expressed, they claim it is recruited to the IRF7 promoter after IFN stimulation. To this regard, may the authors provide more details to support their conclusions. Do the authors hypothesize that information about the modifications MNDA undergoes post-translational changes after IFN stimulation in order to translocate on the IRF7 promoter and activates Pol II recruitment? Do other transcription factors (Sp1!), in addition to MNDA, contribute to Pol II recruitment?

11. Supplementary Figure 6. According to this figure, IRF7 binds the IFN β promoter at 4 hours post treatment, showing a relevant difference between MNDA shRNA cells and Control shRNA. Nevertheless, in fig. 2B and 1D the same model doesn’t show any difference in IFN β expression levels even after longer times. The authors should justify this difference.

In conclusion, the study appears highly premature to be accepted in the present form and it needs a thorough revision before publication in Nature Communications.

Reviewer #2 (Remarks to the Author):

The manuscript reports the interesting observation that the human PYHIN protein MNDA enhances IFN α synthesis through transcriptional control of their regulator IRF7.

Main comments:

1. A big weakness of the manuscript is that it uses shRNA-based, incomplete knockdown of MNDA for the first four main figures and the majority of supplementary figures and for the remaining work cells made MNDA-deficient by CRISPR/Cas9-based knockout (excepting fig. 6C). While results with the two experimental systems are not contradictory for the most part, some of the data with the shRNA system are inconsistent, e.g. the kinetics of DNA-induced IFN α 14 expression and the binding of RNA pol II to its promoter, figs 3B, C, or the lack of effect of MNDA on ISG expression in DNA-transfected cells that are expected to produce less IFN α (figs. S5B, C). Some of the data are of poor technical quality, e.g. western blots S2E, S7A (unclear how these bands could be quantified), 3H (no comment on why MNDA in the shRNA cells disappears after 70-mer transfection). In conclusion, the paper would gain much in clarity and persuasiveness with the complete dataset generated in the cleaner experimental system, i.e. the knockout cells and their reconstituted counterparts.

2. The interpretation of the effect on IFN α synthesis is entirely based on the impact of MNDA on IRF7 synthesis. This should be controlled with IRF7-deficient cells.

3. The paper would be stronger if additional information on the role of MNDA in RNA polymerase II recruitment would be provided. For example, the establishment of activating histone marks, initiation complex assembly, or mediator recruitment could be examined by ChIP.

Technical comments:

1. A reference should be provided for the description of GM-CSF or CSF-1-differentiated monocytes as M1 or M2 polarised. To my knowledge this requires additional polarising stimuli such as IFN γ or IL4.
2. Based on the normalisation procedure described in materials and methods, the y-axes of the ChIP graphs generated with the knockout cells are most likely mislabelled and should read 'fold induction over untreated' or similar. The values are much too high for % input.

RESPONSE TO REVIEWER COMMENTS (original reviewer comment in *italics*)

We thank the reviewers for their constructive comments which have enabled us to strengthen the paper with new data, that further confirms the central role of MNDA in regulation of IRF7-dependent type I interferon, including in primary monocytes, and also provides some further mechanistic insights into how MNDA regulates the IRF7 promoter. New data is presented as Fig 1D, E, K, L; Fig 3B, C, F, G; Fig 5E; Fig 6H, I, J; Fig S6C, E; Fig S8I, Fig S10A, B, C. Changes and additions to the main text and supplementary information are highlighted in yellow.

Reviewer #1 (Remarks to the Author):

In the study entitled "MNDA controls the pathogen-stimulated type I interferon cascade in human monocytes by transcriptional regulation of IRF7" by Casserly et al., the authors show that the PYHIN protein MNDA is required for IFN type I induction in monocytes. Unlike other PYHINs, this was not due to a pathogen sensing role, but rather MNDA regulated expression of IRF7, a transcription factor essential for type I IFN induction. Mechanistically, MNDA was required for recruitment of RNA polymerase II to the IRF7 gene promoter, and in fact MNDA was itself recruited to the IRF7 promoter after type I IFN stimulation. Overall, these data provide evidence that MNDA is a critical regulator of the type I IFN cascade in human myeloid cells. Although this conclusion appears novel and interesting, unfortunately the study suffers from several flaws that should be addressed in order to strength the data and make it suitable for publication.

>We thank the reviewer for recognising that our study is novel and interesting, and below we have addressed the flaws highlighted by the reviewer.

General points:

1.1. MNDA, expressed in myelomonocytic and B lymphoid hematopoietic cells, associates with chromatin, but does not bind specific DNA sequences (Xie J. et al., J Cell Biochem., 1998). MNDA participates in a ternary complex with the transcription factor YY1 target DNA element. According to this may the authors investigate if in their system MNDA binds DNA even at low affinity? The results achieved could shed some light on the mechanisms MNDA relies on to recruit RNA polymerase II on the IRF7 promoter.

>From work in our lab and others, we know that in any dsDNA binding assay, PYHIN proteins will interact with the DNA in a sequence-independent manner, consistent with the known structure of IFI16 HIN domains, which bind to the phosphate backbone of dsDNA¹. Indeed, we can immunoprecipitate MNDA from cells using dsDNA. Therefore, we do not think it would be informative to provide data on MNDA in a dsDNA binding assay. Rather, Fig 7C-E show that Flag-MNDA can ChIP to the IRF7 promoter, and this associated is IFN-dependent (Fig 7E). This interaction between the IRF7 promoter and MNDA is specific, since it is not observed for another IFN α -dependent promoter, IRF1 (Fig 7F). As discussed in the Discussion, what defines the specificity of MNDA binding to the IRF7 promoter is an open question, as is the issue of whether the binding is direct to the DNA, or part of a complex. In the future we hope to use RNA-seq and ChIP-seq to explore this issue. Of note, other PYHIN proteins have also been shown to be recruited to specific gene promoters, since recently IFI16 was shown by ChIP to bind to the RIG-I promoter in response to influenza A infection of cells². New data, discussed below, showing that MNDA is required for STAT2 recruitment to the IRF7 promoter, and that there are more positive chromatin marks (H4 acetylation) on the promoter in the presence of MNDA, do shed some more light on the mechanism used by MNDA to regulate IRF7.

1.2. Does MNDA directly interact with RNA polymerase II and, in positive case, may the authors provide information about the domains involved in such an interaction?

>We are unsure if MNDA interacts directly with RNA polymerase II, but if it did so, it would be difficult to understand the specificity of MNDA for the IRF7 promoter, and not many other Pol II-dependent promoters. Instead, we focused on providing more information on the activation of the IRF7 promoter by MNDA by measuring Sp1, STAT2 and H4-ac binding activity (see below).

1.3. It has been recently demonstrated (Bosso et al., PLOS Pathogens, 2020) that MNDA interacts with some transcription factors, including Sp1, in order to restrict HIV replication in macrophages. On the light of these results, the authors should investigate by Electrophoretic Mobility Shift Assay along with antibody supershift and Chromatin Immunoprecipitation if other transcription factors are recruited or bound on the IRF7 promoter where they might interact with MNDA after type I IFN stimulation. Otherwise, as it is now, a potential reader may get the impression that MNDA is the only transcription factor driving the activity of the IRF7 gene promoter. This cannot be true when dealing with such a complex promoter as that of IRF7.

>To address this issue, we first considered which transcription factors to test. As noted by the reviewer, Bosso et al showed that MNDA can interact with Sp1³, and Sp1 has a role in the induction of many promoters. Therefore, it was possible that MNDA could regulate the IRF7 promoter through an effect on Sp1. However, ChIP analysis of Sp1 binding to the IRF7 promoter showed no difference in the presence or absence of MNDA (new data as Fig S10). There was no difference in the ChIP signal either on the whole promoter region (Fig S10A, B) or in response to an IFN α time course. We then considered other transcription factors and searched the literature to discover what transcription factors had been previously shown to interact with the human IRF7 promoter. Using ChIP-seq, Au-Yeung and Horvath showed that in HeLa cells, STAT2 binding was strongly enriched at the IRF7 promoter after IFN α treatment of cells⁴. We therefore investigated whether IFN α -dependent STAT2 recruitment to the IRF7 promoter in THP-1 cells was MNDA-dependent. We found that STAT2 recruitment was indeed MNDA-dependent, since the fold enrichment of STAT2 binding to the IRF7 promoter was significantly less in cells lacking MNDA compared to cells expressing Flag-MNDA (new data as Fig 6H). Further, in an IFN α time course, significantly less STAT2 was recruited to the IRF7 promoter in the absence of MNDA compared to cells expressing MNDA (new data as Fig 6I). Thus, MNDA is required for the recruitment of a critical STAT transcription factor known to have a role in human IRF7 promoter induction. Also, we analysed the presence of positive histone marks on the IRF7 promoter, and showed that for IFN α -stimulated cells, the ChIP signature for Histone H4K5,8,12,16ac (ac-H4), which is associated with active genes, was significantly reduced in cells lacking MNDA compared to Flag-MNDA expressing cells (new data as Fig 6J). How exactly MNDA regulates STAT2 recruitment and promoter accessibility will be the focus of future work.

Specific points:

1.4. Introduction, line 68. The reference showing that IFI16 restricts the growth of HPV (Lo Cigno et al. J. Virol, 2015) in addition to Herpesviruses and HIV is missing and should be also mentioned in the text.

>We apologise for the omission and have added the statement and reference

1.5. Figure 1C [now Fig 1B], line 113-114. "Since MNDA protein expression was higher in monocytes than macrophages in contrast to IFI16 where the opposite was the case...." This comment does not correspond to the data presented in the Immunoblot showing that both unprimed or PMA-primed THP-1 express comparable levels of the IFI16 protein. The sentence should be modified.

>We have modified the sentence, which should have referred to unstimulated monocytes and macrophages, where the opposite is indeed the case. We have modified the text to say: 'Since MNDA protein expression was higher in unstimulated monocytes than macrophages, in contrast to IFI16 where the opposite was the case'

1.6. Moreover, the authors should explain why the IFN-beta mRNA levels of control shRNA cells upon 70mer stimulation (panel E, >20 fold) are much lower when compared to those reported for the same treatment in the same cells in the kinetics shown in panel D, 100-200 fold. They should be similar.
>The difference is due to day-to-day variability in the cell's response to 70mer, for example due to differing transfection efficiencies. We have replaced both panels with a new experiment where IFI16 and MNDA shRNA cells are compared to control shRNA cells in the same time course, so that the exact same stimulation level can be compared to cells with reduced MNDA and IFI16 expression (new data as Fig 1C).

1.7. In addition, to substantiate their data, the authors should also perform kinetic experiments with IFI16-depleted cells as performed with MNDA-depleted cells. Figure 1E. Kinetics experiments for IFI16 shRNA should be included as in panel D.

>This is included in the new data in Fig 1C where an identical time course is done for IFI16 and MNDA shRNA cells (0, 3, 6, 12, 24h 70mer stimulation).

1.8. Figure 1I-K [now Fig 1H-J]. To provide compelling evidence about IFN-alpha regulation, it is mandatory to quantify the protein in the culture supernatants by ELISA or similar assays.

>We repeated the experiment using MNDA siRNA to knock down MNDA expression in primary human monocytes using six new donors, in order to also collect supernatants for analysis of IFN α secretion. New data in Fig 1H-I shows similar results to the previous submission, in that the siRNA was effective as reducing MNDA expression, and this led to an inhibition of DNA-stimulated IFN α but not IFN β mRNA induction. New data in Fig 1K shows that supernatants from these MNDA siRNA-treated donor cells displayed significantly reduced IFN α secretion. Further, this had a significant effect on the overall bioactivity of type I IFN in the supernatants as shown by a type I IFN bioassay (new data in Fig 1L).

1.9. Figure 3B-C [now Fig S7A, B]. As shown in panel B [A], the IFNalpha14 mRNA is highly inducible at 12 h in THP-1 cells upon stimulation with 70mer. By contrast, robust Pol II recruitment is detectable 16 h after stimulation suggesting that IFNalpha14 mRNA expression precedes its transcription by Pol II. This discrepancy should be explained.

>The discrepancy is likely due to the sensitivity of the assays – IFN α 14 induction reaches 100s of fold above control whereas the Pol II recruitment assay reaches 2-fold above control. Therefore, it is not surprising that an increase in IFN α 14 mRNA is measurable before and increase in Pol II recruitment is detectable. As well as retaining this data for the shRNA as a supplemental figure, we have repeated the ChIP experiment in MNDA KO cells (new data as Fig 3C). This new data shows an even more marked effect of MNDA deficiency on recruitment of Pol II to the IFN α 14 promoter in response to 70mer stimulation, and in that case an increased recruitment of Pol II is seen earlier, at 8h.

1.10. Figure 3H-J [now Fig S8A-C]. "Interestingly, dsDNA-, RNA virus- and type I IFN-stimulated IRF7 protein expression were all diminished in cells with reduced MNDA expression, compared to control cells". This conclusion does not seem to be supported by the included immunoblot. In THP-1 cells expressing control or MNDA shRNA transfected with dsVACV or stimulated with IFN-alpha, IRF7 expression is not significantly decreased at 24 h after stimulation going against the conclusion drawn by the authors. This discrepancy must be justified.

>We disagree that IRF7 protein is not significantly decreased at 24 h but accept that the effect appears more subtle than for VSV stimulation. Therefore, we have added a densitometric analysis of the bands, and included the values for IRF7 intensity under the IRF7 blots, which clearly shows reduced IRF7 at 24 h for dsDNA (Fig S8A) or IFN α (Fig S8C) stimulation. Furthermore, this conclusion is consistent with data in the MNDA KO cells (Fig 4A).

1.11. Figure 3M [now Fig S8I]. The indicated times considered for the analysis should be the same as in figures 3K and 3L [now S8G & H].

>We have repeated the time course so that all three panels are the same, showing 0, 3, 6, 12, 24 h treatment with the stimulants, namely dsRNA (Fig S8G), dsDNA (Fig S8H) and IFN α (Fig S8I).

1.12. Figure 4. The authors should perform the same experiments of supplementary figures 2E and 2F in Cal-1 pDCs shRNA-treated cells to prove the reduction of MNDA's expression. To this purpose, an Immunoblot analysis of MNDA protein expression in CAL-1 pDCs cells stably expressing control or MNDA shRNA, and a quantitative PCR analysis of MNDA mRNA from CAL-1 pDCs cells stably expressing control or MNDA shRNA should be performed.

>We had already included data showing that MNDA shRNA reduces MNDA mRNA in CAL-1 cells in the previous submission (old Fig 4A). This is shown in the revised submission in Fig 5A. As requested, we have also analysed MNDA protein expression by immunoblot to show an effect of the MNDA shRNA on MNDA protein. Because MNDA protein expression is lower in CAL-1 pDCs than THP-1 monocytes, we had to first IP MNDA and then immunoblot for it. New data in Fig 5E clearly shows that MNDA protein expression is reduced in cells treated with MNDA shRNA compared to Control shRNA, since less MNDA is immunoprecipitated from the former compared to the latter. We have added details of the IP in the Methods section.

1.13. Figure 5 [now Fig 4]. To confirm the removal of MNDA genetic locus with the CRISPR/Cas9 technology, the authors should perform a TIDE analysis.

>We did a TIDE analysis, which showed disruption of the MNDA locus. But we also performed NGS which is more compelling, and the original submission showed the sequences obtained from that analysis and their percent composition. We have now added to this the indel length histogram analysis to provide a further visual demonstration of the disrupted MNDA locus (new data as Fig S6C and E).

1.14. Figures 6-7. In this set of experiments the authors demonstrate that IFN- α -dependent Pol II recruitment to the IRF7 promoter is MNDA-dependent. Although they show that MNDA is constitutively expressed, they claim it is recruited to the IRF7 promoter after IFN stimulation. To this regard, may the authors provide more details to support their conclusions. Do the authors hypothesize that information about the modifications MNDA undergoes post-translational changes after IFN stimulation in order to translocate on the IRF7 promoter and activates Pol II recruitment?

>It is currently unclear how IFN α regulates MNDA. For IFI16, cellular stimulation can cause re-distribution of IFI16 from the nucleus to the cytoplasm, due to acetylation its NLS. But it is unclear whether IFN α treatment of cells causes PTM of MNDA. In contrast to IFI16, to our knowledge MNDA resides solely in the nucleus, consistent with its role in transcription. Although it is currently unclear how MNDA recruitment to the IRF7 promoter is regulated by IFN α , the new data presented here for STAT2 and H4-ac are consistent with an important role for MNDA in modulating promoter accessibility and recruitment of known transcription factors.

1.15. Do other transcription factors (Sp1!), in addition to MNDA, contribute to Pol II recruitment?

>See response to 1.3 above

1.16. Supplementary Figure 6 [now Fig S7]. According to this figure, IRF7 binds the IFN β promoter at 4 hours post treatment, showing a relevant difference between MNDA shRNA cells and Control shRNA. Nevertheless, in fig. 2B and 1D [now Fig 1C] the same model doesn't show any difference in IFN β expression levels even after longer times. The authors should justify this difference.

>As the reviewer notes, clearly IRF7 is recruited to the IFN β promoter at 4 h (Fig S7E after DNA stimulation). And since MNDA is required for IRF7 expression, MNDA shRNA prevents this

recruitment (Fig S7E). But as the reviewer notes, MNDA shRNA does not have an effect on IFN β mRNA induction in response to up to 12 h direct DNA stimulation (Fig 1C) or up to 24 h poly(I:C) stimulation (Fig 2B). This must mean that although IRF7 can be recruited to the IFN β promoter, it does not have a role in IFN β promoter induction until later times during the IFN I cascade (Fig 3A). Consistent with this hypothesis, in the siRNA experiment shown in Fig S3C, there is a small but significant effect of MNDA siRNA on poly(I:C)-stimulated IFN β at 24 h but not at 12 h. Also, in Fig S7H, priming cells with IFN α , which would increase IRF7 expression, renders IFN β induction more sensitive to MNDA, since the increased induction of IFN β due to priming is abrogated in cells expressing MNDA shRNA.

Reviewer #2 (Remarks to the Author):

The manuscript reports the interesting observation that the human PYHIN protein MNDA enhances IFN α synthesis through transcriptional control of their regulator IRF7.

Main comments:

2.1. A big weakness of the manuscript is that it uses shRNA-based, incomplete knockdown of MNDA for the first four main figures and the majority of supplementary figures and for the remaining work cells made MNDA-deficient by CRISPR/Cas9-based knockout (excepting fig. 6C). While results with the two experimental systems are not contradictory for the most part, some of the data with the shRNA system are inconsistent, e.g., the kinetics of DNA-induced IFN α ₁₄ expression and the binding of RNA pol II to its promoter, figs 3B, C, [now Fig S7A, B] or the lack of effect of MNDA on ISG expression in DNA-transfected cells that are expected to produce less IFN α (figs. S5B, C). Some of the data are of poor technical quality, e.g. western blots S2E, S7A [now Fig S8D] (unclear how these bands could be quantified), 3H [now S8A] (no comment on why MNDA in the shRNA cells disappears after 70-mer transfection). In conclusion, the paper would gain much in clarity and persuasiveness with the complete dataset generated in the cleaner experimental system, i.e. the knockout cells and their reconstituted counterparts.

>In working with the PYHINs, we have sometimes discovered differences in results using RNAi versus CRISPR KO, or in the case of mouse PYHINs, differences in shRNA experiments compared to cells from KO mice. This is in part due to different family members being able to compensate for complete or partial loss of another PYHIN. Therefore, we feel it is important to have data with both knock-down and knock-out of MNDA in this study, and it is reassuring that the data with shRNA versus siRNA versus CRISPR/Cas9 KO is largely consistent. However, in response to the reviewers concerns we have significantly re-organised the manuscript, to give the CRISPR KO data more prominence, and have also repeated many of the key observations in shRNA cells in CRISPR KO cells. Specific changes in the manuscript, and responses to the specific issues raised above are:

- The CRISPR KO cell data is now introduced earlier, in Fig 3.
- Much of the shRNA data has now been moved to supplemental figures
- New CRISPR KO data is presented as Fig 3B (time course for IFN α induction in response to dsDNA stimulation), Fig 3C (ChIP analysis of IFN α ₁₄ promoter), Fig 3F & 3G (in response to point 2.2 below), Fig 6H, I, J (new ChIP data showing a role for MNDA in STAT2 recruitment and effect of MNDA on H4-ac chromatin marks), Fig S10A, B, C (ChIP analysis of Sp1 binding to IRF7 promoter).
- In terms the kinetics of DNA-induced IFN α ₁₄ expression and the binding of RNA pol II to its promoter, see response to point 1.9 above.
- In terms of the lack of an effect of MNDA on ISG expression in DNA-transfected cells (figs. S5B, C), we would conclude that these responses are mainly IFN β -dependent. This result is not inconsistent with any of the MNDA KO data. For both shRNA and KO cells we see an effect on DNA-stimulated IRF7 mRNA. And in fact, the only other DNA-stimulated impairments in gene induction seen are for three strongly IRF7-dependent gene induction events: IFN α (for both shRNA and KO cells), IFNL2 (for KO cells) and IFN α -primed IFN β (for

shRNA cells). No effect was seen on DNA-stimulated IFIT1 or IFIT2 mRNA or IFIT3 protein (shRNA) nor DNA-stimulated IFNL1 (KO).

- Western blot previously presented as Fig S2E has been removed as Fig 1D shows a more comprehensive analysis of MNDA protein knock down in shRNA cells.
- For the quantitative analysis of the bands in Fig S8D, we have added further explanation in the legend and Methods as to how this was done.
- For the MNDA blot in Fig S8A, it is possible that the MNDA shRNA is slightly more effective in the presence of 70mer, but this does not affect the interpretation of the data in that at 24h of 70mer treatment, clearly there is a significant difference both between MNDA protein expression and IRF7 protein expression in the MNDA shRNA cells compared to control shRNA cells.

2.2. *The interpretation of the effect on IFN α synthesis is entirely based on the impact of MNDA on IRF7 synthesis. This should be controlled with IRF7-deficient cells.*

>We addressed this by using IRF7 siRNA in MNDA KO cells. The new data is shown in Fig 3F & G and described in the Results section as follows: 'In order to confirm that MNDA-dependent IFN α induction was via an effect on IRF7, we used IRF7 siRNA in WT and *MNDA*^{-/-} cells to examine whether IRF7-dependent IFN α induction remained in the absence of MNDA. Fig. 3F shows that IRF7 siRNA treatment of WT cells effectively suppressed IRF7 induction by DNA. IRF7 siRNA also inhibited DNA-stimulated IFN α induction, and to a similar degree to gene ablation of MNDA (Fig. 3G). Compellingly, in *MNDA*^{-/-} cells, IRF7 siRNA had no further effect on IFN α induction (Fig. 3G).'

2.3. *The paper would be stronger if additional information on the role of MNDA in RNA polymerase II recruitment would be provided. For example, the establishment of activating histone marks, initiation complex assembly, or mediator recruitment could be examined by ChIP.*

>See response to point 1.3 above – we have now shown a role for MNDA in STAT2 recruitment and in the appearance of activating histone marks (H4-ac) at the IRF7 promoter.

Technical comments:

2.4. *A reference should be provided for the description of GM-CSF or CSF-1-differentiated monocytes as M1 or M2 polarised. To my knowledge this requires additional polarising stimuli such as IFN γ or IL4.*

>The reviewer's point is well taken. What exactly GM-CSF- and M-CSF-differentiated monocytes are called is controversial. Here we simply wanted to differentiate monocytes into two different macrophage lineages. We have modified the text to not mention M1 and M2, and also provided a reference that characterises the differences between GM-CSF- and M-CSF-differentiated monocytes: Immunoblot analysis confirmed strong expression of MNDA in monocytes, which was reduced upon differentiation of cells into different macrophage lineages by treatment of monocytes with either GM-CSF or M-CSF⁵

2.5. *Based on the normalisation procedure described in materials and methods, the y-axes of the ChIP graphs generated with the knockout cells are most likely mislabelled and should read 'fold induction over untreated' or similar. The values are much too high for % input.*

>The reviewer is correct. We have corrected the mislabel and all ChIP data is now presented as 'Fold enrichment'. The Methods state that this refers to fold enrichment relative to the untreated control. In the case of Sp1 (new data in Fig S10) we present the data as both % of input and 'fold enrichment' to illustrate that Sp1 constitutive occupancy of the IRF7 promoter is not affected by MNDA, and neither is Sp1 recruited to the IRF7 promoter after IFN α treatment.

References:

- 1 Jin, T. *et al.* Structures of the HIN domain:DNA complexes reveal ligand binding and activation mechanisms of the AIM2 inflammasome and IFI16 receptor. *Immunity* **36**, 561-571, doi:10.1016/j.immuni.2012.02.014 (2012).
- 2 Jiang, Z. *et al.* IFI16 directly senses viral RNA and enhances RIG-I transcription and activation to restrict influenza virus infection. *Nat Microbiol* **6**, 932-945, doi:10.1038/s41564-021-00907-x (2021).
- 3 Bosso, M. *et al.* Nuclear PYHIN proteins target the host transcription factor Sp1 thereby restricting HIV-1 in human macrophages and CD4+ T cells. *PLoS pathogens* **16**, e1008752, doi:10.1371/journal.ppat.1008752 (2020).
- 4 Au-Yeung, N. & Horvath, C. M. Histone H2A.Z Suppression of Interferon-Stimulated Transcription and Antiviral Immunity Is Modulated by GCN5 and BRD2. *iScience* **6**, 68-82, doi:10.1016/j.isci.2018.07.013 (2018).
- 5 Lacey, D. C. *et al.* Defining GM-CSF- and macrophage-CSF-dependent macrophage responses by in vitro models. *J Immunol* **188**, 5752-5765, doi:10.4049/jimmunol.1103426 (2012).

REVIEWER COMMENTS

Reviewer #1 (Remarks to the Author):

Prof. Bowie and colleagues have revised their manuscript "MnDA controls the pathogen-stimulated type I interferon cascade in human monocytes by transcriptional regulation of IRF7", NCOMMS-20-48298A, resubmitted to Nature Communications. They have added new data in response to reviewers' concerns that provides additional evidence and confidence in authors' interpretation. The manuscript now objectively opens a new area of investigation and it is ready for publication in Nature Communications.

Reviewer #2 (Remarks to the Author):

The revised manuscript has addressed my concerns with an improved structure and important new data. Regarding the new figure demonstrating MnDA-regulated binding of Stat2 the authors should discuss the Stat2 binding sites they identified with regard to published ISRE sequences in the IRF7 promoter and 5' UTR. They should further discuss whether the data support a direct interaction of MnDA and IRF7 at promoter level or whether they rather suggest a functional cooperation without physical contact.

Response to reviewers

In terms of the final reviewers comments, Reviewer #2 requested that *'Regarding the new figure demonstrating Mnda-regulated binding of Stat2 the authors should discuss the Stat2 binding sites they identified with regard to published ISRE sequences in the IRF7 promoter and 5' UTR. They should further discuss whether the data support a direct interaction of MNDA and IRF7 at promoter level or whether they rather suggest a functional cooperation without physical contact.'*

To address this we have added the following sentences to the discussion:

"After IFN α stimulation, we detected STAT2 binding to the IRF7 promoter at regions known to contain the IFN-stimulatory response element (ISRE) that STAT2 binds to, namely upstream of the TSS near position -250, and downstream of the TSS in the 5'UTR near position +250^{47, 48}. We also detected a strong peak of binding further downstream between -500 to -384. In all cases, STAT2 binding was partially or fully dependent on the presence of MNDA."

And

"Also, it is unclear whether MNDA physically associates directly with the IRF7 promoter, or is part of a larger protein complex that regulates promoter accessibility."